# DOES GENERATIVE RETRIEVAL BREAK THROUGH THE LIMITATIONS OF DENSE RETRIEVAL?

## ABSTRACT

Generative retrieval (GR) has emerged as a new paradigm in neural information retrieval, offering an alternative to dense retrieval (DR) by directly generating identifiers of relevant documents. In this paper, we theoretically and empirically investigate how GR fundamentally diverges from DR in both learning objectives and representational capacity. GR performs globally normalized maximum-likelihood optimization and encodes corpus and relevance information directly in the model parameters, whereas DR adopts locally normalized objectives and represents the corpus with external embeddings before computing similarity via a bilinear interaction. Our analysis suggests that, under scaling, GR can overcome the inherent limitations of DR, yielding two major benefits. First, with larger corpora, GR avoids the sharp performance degradation caused by the optimization drift induced by DR's local normalization. Second, with larger models, GR's representational capacity scales with parameter size, unconstrained by the global low-rank structure that limits DR. We validate these theoretical insights through controlled experiments on the Natural Questions and MS MARCO datasets, across varying negative sampling strategies, embedding dimensions, and model scales. But despite its theoretical advantages, GR does not universally outperform DR in practice. We outline directions to bridge the gap between GR's theoretical potential and practical performance, providing guidance for future research in scalable and robust generative retrieval.

## 1 INTRODUCTION

Advances in deep learning and representation learning (Vaswani et al., 2017; Lee & Toutanova, 2018) have established neural information retrieval (IR) as the dominant paradigm (Mitra et al., 2018; Fan et al., 2022). Within this paradigm, dense retrieval (DR) encodes queries and documents into vectors and measures their similarity through bilinear interactions, enabling efficient vectorized recall and delivering state-of-the-art performance across diverse retrieval tasks (Karpukhin et al., 2020; Khattab & Zaharia, 2020). Recently, driven by generative large language models (LLMs) (Radford et al., 2018; Yang et al., 2025b; Lewis et al., 2020), generative retrieval (GR) has emerged as a new branch of neural IR (Tay et al., 2022; Bevilacqua et al., 2022; Zhuang et al., 2022; Wang et al., 2022; Li et al., 2024; Zeng et al., 2024b). GR directly generates identifiers of relevant documents (docids) for a given query, with corpus knowledge embedded in the model parameters. It typically adopts a sequence-to-sequence architecture trained with cross-entropy loss, while inference relies on constrained decoding to ensure valid docids.

To better understand GR, recent studies have examined its connection to DR. Some research interprets GR as implicitly performing dot-product scoring within an LLM's parameters and propose a unified framework for similarity computation across both paradigms (Nguyen & Yates, 2023; Wu et al., 2024). Despite this formal unification, the substantial differences in model architecture should not be overlooked: DR is encoder-only, whereas GR employs an autoregressive model with a decoder. This naturally raises the question:

*Do GR and DR differ fundamentally in their modeling mechanisms for retrieval?*

We address this question along two dimensions: (i) *Learning objective*: DR trains with local normalization over a small candidate set in document space, whereas GR maps the problem to vocabulary space and optimizes a globally normalized likelihood; and (ii) *Representational capacity*:

DR encodes queries and documents as low-dimensional embeddings, while GR uses the full model parameters to memorize the entire corpus.

Our **theoretical analysis** elaborates on these aspects and leads to the following conclusions: DR has intrinsic bottlenecks in both learning and representation that constrain its performance under scaling of corpus and model size, whereas GR does not. First, local normalization in DR introduces calibration errors that grow with corpus size, whereas GR's global normalization avoids such optimization drift and benefits more from larger corpora. Second, the low-rank constraint imposed by DR's embedding dimension limits its ability to approximate the (often higher-rank) true query-document relevance matrix, whereas GR's parameterization allows higher-rank approximations, making it better suited to leverage large-scale models.

To validate our theoretical analysis **empirically**, we evaluate standard DR, multi-vector DR (MVDR) (Khattab & Zaharia, 2020; Formal et al., 2021; Li et al., 2023a) and two GR variants following the DSI (Tay et al., 2022) framework on the Natural Questions (NQ) (Kwiatkowski et al., 2019) and MS MARCO (Bajaj et al., 2016) datasets. Under controlled settings, we conduct three studies: (i) By varying DR's negative sampling and embedding dimension, we evaluate their effects on calibration error and ranking metrics; experimental results show optimization limits due to local normalization and representation limits due to the embedding dimension. (ii) By scaling GR and DR with matched model sizes and training corpus sizes, we observe larger gains for GR, providing preliminary evidence that GR has the potential to overcome DR's bottlenecks when scaled. (iii) Using a larger model with 14B parameters, we conduct zero-shot and test-time scaling experiments for GR and observe promising performance, further supporting the scaling advantages that GR may obtain.

Overall, our theoretical and empirical results highlight key modeling differences between GR and DR, showing that GR avoids DR's bottlenecks and has greater potential as an IR paradigm at larger data and model scales. However, our experiments are limited to in-distribution queries, and neither the model nor the data scale is arbitrarily large. In practice, GR does not consistently outperform DR, as its effectiveness depends on factors such as docid design (Bevilacqua et al., 2022; Li et al., 2023b), training data construction (Zhuang et al., 2022), and decoding strategies (Zeng et al., 2024a; Lee et al., 2022). We conclude by discussing these limitations and outlining future directions to close the gap between GR's theoretical promise and practical performance.

## 2 PRELIMINARIES

**Problem statement.** Let $\mathcal{Q}$ be a set of queries and $\mathcal{D} = d_1, \ldots, d_N$ a document collection. Let $P^\star(d \mid q)$ denote the unknown ground-truth conditional distribution of documents given query $q$. Training pairs $(q, d^+)$ are drawn from a data distribution $\mathcal{D}_{\text{train}}$, where $d^+$ is a relevant document under $P^\star(\cdot \mid q)$. The goal of IR is to approximate $P^\star(d \mid q)$ using a parametric model $P_\Theta(d \mid q)$, ensuring both probabilistic calibration and high ranking quality (Chowdhury, 2010).

**Dense retrieval.** Let $e_q \in \mathbb{R}^r$ and $e_d \in \mathbb{R}^r$ denote the query and document embeddings from encoders $f_q$ and $f_d$, respectively (Karpukhin et al., 2020; Xiong et al., 2020a). The DR score for a pair is computed as their inner product $S(q, d) = e_q^\top e_d$, and the locally normalized (e.g., in-batch) softmax loss is defined accordingly:

$$P_\Theta(d \mid q; \mathcal{N}) = \frac{\exp(S(q, d)/\tau)}{\sum_{d' \in \{d\} \cup \mathcal{N}(q)} \exp(S(q, d')/\tau)}, \tag{1}$$

where $\mathcal{N}(q)$ is the negative set and $\tau > 0$ is a temperature. The standard contrastive objective is:

$$\mathcal{L}_{\text{DR}}(\Theta) = \mathbb{E}_q\big[-\log P_\Theta(d^+ \mid q; \mathcal{N}(q))\big]. \tag{2}$$

Eq. 2 encourages $S(q, d^+)$ to exceed the scores of negatives within the current candidate pool. In practice, negatives may come from the in-batch sampling (Karpukhin et al., 2020; Khattab & Zaharia, 2020) or hard-negative mining (Xiong et al., 2020a; Zhan et al., 2021).

**Generative retrieval.** Each document has a tokenized docid $y_{1:L} \in \mathcal{V}^L$ from a finite vocabulary $\mathcal{V}$ (Tay et al., 2022). The GR training loss is defined by a sequence generation model $p_\Theta(y_t \mid y_{<t}, q)$:

$$\mathcal{L}_{\text{GR}}(\Theta) = \mathbb{E}_q\big[-\log P_\Theta(d^+ \mid q)\big] = \mathbb{E}_q\Big[-\sum_{t=1}^{L} \log p_\Theta\big(y_t^+ \mid y_{<t}^+, q\big)\Big]. \tag{3}$$

The mapping between sequences in $\mathcal{V}^L$ and $\mathcal{D}$ is constrained, so that decoding a sequence deterministically selects a document. At inference time, beam search is used with prefix constraints (e.g., trie) to guarantee valid docids.

# 3 THEORETICAL ANALYSIS

## 3.1 LEARNING OBJECTIVES

Here, we refer to an objective as *local* when normalization is restricted to the sampled candidate set, whereas a *global* objective normalizes over the entire document collection $\mathcal{D}$. §3.1.1 presents DR's locally normalized surrogate and formalizes the resulting calibration gap, while §3.1.2 then shows that GR optimizes a globally normalized likelihood objective.

### 3.1.1 DR LOCALLY NORMALIZES SURROGATE

The DR objective in Eq. 2 minimizes a surrogate defined on the set $\{d^+\} \cup \mathcal{N}(q)$, renormalizing scores via a softmax within $K$ candidates per batch. This makes the learning objective explicitly dependent on the sampled negatives, implying that the negative-sampling scheme (both the size of the candidate set and the quality of the negatives) has a substantial impact on the final performance of DR. Ideally, one would use as negatives the entire set of non-relevant documents, but this is computationally infeasible under realistic resource constraints (Wang & Isola, 2020). This mismatch leads to a calibration gap between the global and local objectives.

*Assumptions.* Negatives for each query $q$ are drawn i.i.d. from a proposal sample policy $\pi(\cdot)$ over $\mathcal{D}$ (with $\mu(\cdot)$ the random sample policy) and scores are bounded as $|S(q,d)/\tau| \leq M$. We define the proposal-bias term

$$\delta(q) = \log \mathbb{E}_{d \sim \pi}\big[e^{S(q,d)/\tau}\big] - \log \mathbb{E}_{d \sim \mu}\big[e^{S(q,d)/\tau}\big]. \tag{4}$$

**Theorem 3.1** (Lower bound under local normalization)**.** *Let $\widetilde{P}_\Theta(d \mid q)$ be the full-softmax distribution. Under the assumptions above, the expected gap satisfies the following condition:*

$$\mathbb{E}_q\Big[\log \widetilde{P}_\Theta(d^+ \mid q) - \log P_\Theta(d^+ \mid q; \mathcal{N}(q))\Big] \geq \log \frac{N}{K} - \mathbb{E}_q[\delta(q)], \tag{5}$$

*where $N = |\mathcal{D}|$ and $K$ is the batch candidate size.*

The proof in Appendix B exposes the mechanism: local normalization replaces the global partition function $Z(q)$ with a batch-level $Z_K(q)$ and, in expectation, $Z_K(q) \approx (K/N)\,Z(q)$ up to proposal bias, yielding a gap that shrinks only logarithmically in $K$, where $Z(q) = \sum_{d'} \exp(S(q,d')/\tau)$ and $Z_K(q) = \sum_{d' \in \{d^+\} \cup \mathcal{N}(q)} \exp(S(q,d')/\tau)$. And a high-probability tail bound version of this theorem is provided in Appendix E.

**Practical mitigations for the calibration gap.** Increasing $K$ and mining harder negatives can partially reduce the gap by better approximating the global normalization, and temperature scaling or post-hoc calibration further helps align scores (Xiong et al., 2020a; Zhan et al., 2021). Nevertheless, as the corpus size $N$ grows, the $\log(N/K)$ term dominates unless $K$ scales proportionally with $N$, making it increasingly hard for DR to match the true posterior calibration.

### 3.1.2 GR FULLY NORMALIZES MAXIMUM LIKELIHOOD

The GR loss in Eq. 3 is the token-level negative log-likelihood of a fully normalized sequence model over docids. Averaging over tokens and queries, the cross-entropy decomposes as

$$\underbrace{\mathbb{E}_q\big[-\log P_\Theta(d^+ \mid q)\big]}_{\text{CE loss}} = \underbrace{\mathbb{E}_q\big[H(P^\star(\cdot \mid q))\big]}_{\text{entropy term}} + \underbrace{\mathbb{E}_q\Big[\mathrm{KL}\big(P^\star(\cdot \mid q) \,\|\, P_\Theta(\cdot \mid q)\big)\Big]}_{\text{KL divergence}}. \tag{6}$$

From the CE–KL decomposition in Eq. 6, the entropy term is constant with respect to the model parameters $\Theta$. We therefore obtain the following proposition, for which a detailed proof is provided in Appendix A:

**Proposition 3.2** (Global normalization and calibration of GR). *Minimizing the GR loss in Eq. 3 is equivalent to minimizing the expected KL divergence in Eq. 6. Consequently, GR permits error-free approximation of the true posterior $P^\star(d \mid q)$ and its objective is equivalent to likelihood-consistent optimization over the globally normalized candidate space.*

Note that teacher forcing makes gradients local to each conditional step, yet the objective itself remains globally normalized. Therefore, even under prefix constraints on the valid code space, improvements in likelihood translate directly into better probability calibration of $P_\Theta(d \mid q)$.

**GR is expected to benefit under corpus scaling.** Based on the above analysis, we conclude that under the assumptions in §3.1.1 for locally normalized DR (fixed negative-sample budget $K$ and proposal bias $\delta(q)$) the gap between the ideal global partition $Z(q)$ and its sampled counterpart $Z_K(q)$ grows with $\log N$ when $K$ and $\delta$ are not increased along with the corpus growth. In practice, this typically manifests as saturation or degradation in retrieval metrics unless $K$ is increased or the sample quality is improved. In contrast, GR optimizes a globally normalized likelihood over the docid space. Assuming a fixed docid scheme with adequate coverage and in-distribution queries, GR does not incur the $\log N$ calibration drift and can keep benefiting from larger corpora without increasing $K$ (albeit with higher computational costs).

## 3.2 REPRESENTATIONAL CAPACITY

§3.2.1 below shows that DR compresses relevance into rank-$r$ structures, inducing a low-rank bottleneck on the relevance matrix, while §3.2.2 shows that GR can approximate the query-document posterior arbitrarily well using its full parameterization.

### 3.2.1 DR EXHIBITS A LOW-RANK BOTTLENECK IN RELEVANCE REPRESENTATION

DR learns a text-to-embedding mapping and computes relevance through a fixed post-interaction rule, typically a bilinear score such as the inner product $S(q, d) = e_q^\top e_d$. Consequently, all relevance information for a query or a document is compressed into an $r$-dimensional vector (Weller et al., 2025). Formally, DR stacks $m$ query embeddings into $Q \in \mathbb{R}^{m \times r}$ and $N$ document embeddings into $D \in \mathbb{R}^{N \times r}$. The resulting relevance matrix is $S = QD^\top \in \mathbb{R}^{m \times N}$, which satisfies $\mathrm{rank}(S) \leq r$ regardless of the encoder architecture, as long as the final interaction is bilinear.

By the Eckart-Young-Mirsky theorem (Eckart & Young, 1936; Mirsky, 1960), among all matrices of rank at most $r$, the truncated SVD of any target logit matrix $S^\star$ achieves the best Frobenius-norm approximation, with minimal error equal to the sum of squared discarded singular values. We therefore state the following corollary:

**Corollary 3.3** (Low-rank bottleneck of bilinear DR). *Let $r$ be the embedding dimension. Any bilinear DR with score $S(q, d) = e_q^\top e_d$ induces a relevance matrix $S = QD^\top$ with $\mathrm{rank}(S) \leq r$. Moreover, for a target $S^\star$, the optimal rank-$r$ approximation error equals the squared singular-value tail $\sum_{i>r} \sigma_i(S^\star)^2$.*

Whenever $S^\star$ exhibits a heavy spectral tail, a fixed-$r$ DR model inevitably suffers from an irreducible approximation error unless $r$ is increased. Contemporaneous work (Weller et al., 2025) also identifies this limitation of DR, providing detailed proofs and experiments, and argues that late-interaction MVDR models (e.g., ColBERT (Khattab & Zaharia, 2020)) may mitigate the issue. However, we show that MVDR remains subject to a similar upper bound when tokens are grouped into channels (see Appendix D for details).

### 3.2.2 GR DIRECTLY FITS THE QUERY-DOCUMENT RELEVANCE MAPPING

Let $\mathcal{V}^L$ denote the docid space with a fixed bijection to documents. GR directly fits the query-document relevance mapping through its full set of model parameters.

**Theorem 3.4** (Approximation of $P^\star$ by GR). *For any $\epsilon > 0$ and any conditional distribution $P^\star(\cdot \mid q)$ supported on $\mathcal{D}$, there exist $L$ and a decoder parameterization such that the induced GR model satisfies $\mathbb{E}_q\big[\mathrm{TV}(P^\star(\cdot \mid q), P_\Theta(\cdot \mid q))\big] < \epsilon$, where $\mathrm{TV}$ denotes the total variation distance.*

Theorem 3.4 states that under a fixed bijective docid coding and for in-distribution queries, a sufficiently expressive GR model can approximate the true query–document relevance mapping arbi-

trarily well (in expected total-variation distance). In other words, with adequate capacity, GR could represents documents, queries, and their relevance relations within the model itself. Note that Theorem 3.4 continues to hold when GR decodes under prefix-constrained decoding (see Appendix C for a detailed proof). Nevertheless, in practice the degree to which GR fits the query-document mapping is affected by several factors, including the quality of the docid tree design and the sufficiency and cleanliness of training data (Tay et al., 2022; Zhuang et al., 2022; Wang et al., 2022). Therefore, Theorem 3.4 is a capacity statement rather than a claim about sample or compute efficiency. It assumes an in-distribution query law and a fixed docid. A highly unbalanced or semantically incoherent docid trie can increase optimization difficulty even under universality, and no guarantee is made for out-of-distribution queries.

**GR is expected to benefit under model scaling.** Under the representation analysis in §3.2, GR can reduce the posterior approximation error by scaling its model capacity (given a fixed docid scheme), whereas DR with bilinear interactions is constrained by an effective rank bound $\mathrm{rank}(S) \leq r$ (or $\leq cr$ with $c$ independent interaction channels). Hence, matching a heavy spectral tail requires proportionally increasing $r$ or $c$. This predicts steeper gains for GR under equal-parameter scaling.

## 4 EXPERIMENTS

We present: (i) experiments that evaluate the theoretical limitations of DR, (ii) synchronized scaling experiments comparing GR and DR, and (iii) experiments that investigate the potential scaling advantages of GR.

### 4.1 EXPERIMENTAL SETUP

We evaluate on two widely used retrieval benchmarks: (i) *Natural Questions (NQ)* (Kwiatkowski et al., 2019): Real user questions paired with supporting evidence from Wikipedia; and (ii) *MS MARCO Passage* (Bajaj et al., 2016): Web search queries from Bing with associated relevant passages. We report the calibration metric *Brier*, computed as the mean squared error between the predicted relevance probability for the top-1 candidate and the ground truth for each query. We also report three retrieval metrics: (i) *Hits@$k$*, (ii) *NDCG@$k$*, and (iii) *MRR@$k$*.

We implement representative systems for DR and GR, deliberately avoiding sophisticated variants to ensure fairness and transparency. For DR, we use: (i) *standard dual encoder* with inner-product scoring (referred to as *Standard DR*), following DPR (Karpukhin et al., 2020); and (ii) a *multi-vector late-interaction* variant (referred to as *MVDR*) in the style of ColBERT-v1 (Khattab & Zaharia, 2020). For GR, we adopt two docid designs and follow a DSI-style training/inference pipeline (Tay et al., 2022): (i) *codebook docids* constructed via residual quantization, where each docid is a length-6 sequence of 8-bit code indices (referred to as *GR-codebook*); and (ii) *text docids* that directly use the document title as the identifier (referred to as *GR-text*). All GR decoding is prefix-constrained by a trie built from the set of valid docids.

To control for capacity and pretraining, all DR models are built on `Qwen3-Embedding-0.6B`, and all GR models use `Qwen3-0.6B` (Yang et al., 2025a). **Full details of the experimental setup are provided in Appendix F, and the implementation details for each subsequent experiment are given in Appendix G.**

### 4.2 LIMITATIONS OF DR

**Optimization limitations introduced by local normalization.** To evaluate the effect of local normalization in DR, we fix all other settings and vary only the number of negative samples and the proportion of hard negatives, and then observe the resulting performance changes.

Figure 1 shows how DR performance changes as the number of negative samples increases. We observe that (i) the calibration metric Brier and the ranking metrics move in tandem, indicating that the theoretically predicted calibration drift manifests as changes in retrieval performance; (ii) all retrieval metrics improve as the number $K$ of negative samples increases and have not plateaued within our compute budget; and (iii) despite a few outliers, Standard DR and MVDR exhibit broadly consistent trends across both datasets.

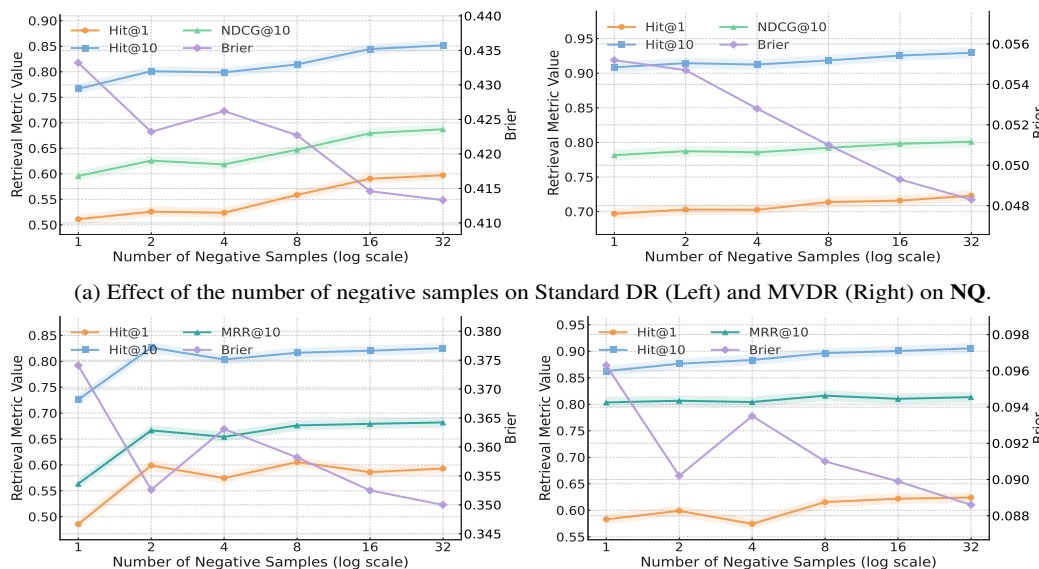

(a) Effect of the number of negative samples on Standard DR (Left) and MVDR (Right) on **NQ**.

(b) Effect of the number of negative samples on Standard DR (Left) and MVDR (Right) on **MS MARCO**.

Figure 1: DR's retrieval performance improves as the number of negative samples increases. The left $y$-axis shows retrieval metrics (higher is better), while the right $y$-axis shows the Brier score (lower is better). The plotted Brier values are raw and thus not comparable across different settings.

Table 1 shows the effect of the *negative-sampling strategy*, showing that DR is highly sensitive to how negatives are chosen. For example, when hard negatives constitute one half of the batch, Standard DR's Hit@1 drops by about 13% relative to using no hard negatives, whereas MVDR's Hit@1 actually improves when mixing in $1/4$ hard negatives. These findings further corroborate the bias introduced by local normalization and indicate that mitigating this limitation purely via negative-sampling heuristics (e.g., injecting hard negatives) is nontrivial.

Table 1: Effect of the hard-negative ratio on DR and MVDR on the NQ dataset.

| Hard-negative ratio | Standard DR | | | MVDR | | |
|---|---|---|---|---|---|---|
| | Hit | | NDCG | Hit | | NDCG |
| | @1 | @10 | @10 | @1 | @10 | @10 |
| 0 | 52.4 | 79.9 | 61.9 | 57.5 | 80.4 | 61.9 |
| 0.25 | 45.4 | 70.3 | 53.0 | 58.4 | 82.4 | 53.0 |
| 0.5 | 39.5 | 63.2 | 46.7 | 52.2 | 78.8 | 46.7 |
| 0.75 | 43.0 | 66.8 | 50.2 | 60.0 | 83.5 | 50.2 |
| 1.0 | 47.0 | 73.8 | 52.2 | 55.6 | 81.6 | 50.2 |

**Representational limitations imposed by embedding dimensionality.** To assess the limitations under bilinear interactions in DR, we vary the embedding size experimentally. Specifically, we append a two-layer non-linear projection after the original output layer to obtain the target embedding dimension, and train this projection jointly with the backbone.

The relationship between embedding dimensionality and DR performance is shown in Figure 2. We observe that: (i) the calibration metric and the ranking metrics vary consistently, indicating that the theoretical effect translates directly into retrieval outcomes; (ii) increasing the embedding dimension yields substantial improvements for both Standard DR and MVDR across datasets, with Standard DR achieving gains of over 20% on the NQ and MS MARCO datasets; and (iii) even at 1024 dimensions, well above the commonly used 768, retrieval performance continues to improve on nearly all curves. Since our datasets are much smaller than real-world corpora, these findings suggest that embedding dimensionality can act as a genuine bottleneck for dimensionality reduction.

### 4.3 SCALING TRENDS OF GR AND DR

**GR and DR under corpus scaling.** To assess how normalization schemes affect corpus-level scaling, we compare GR and DR on progressively larger corpora. We sample document and query subsets of varying sizes from the official training and evaluation sets, and train/evaluate GR and DR on matched subset sizes. All hyperparameters are held fixed except corpus size. To isolate training budget effects, we keep it fixed and vary only the number of candidate documents, increasing it logarithmically from a base equal to the number of documents seen during training (300K for NQ and 1M for MS MARCO).

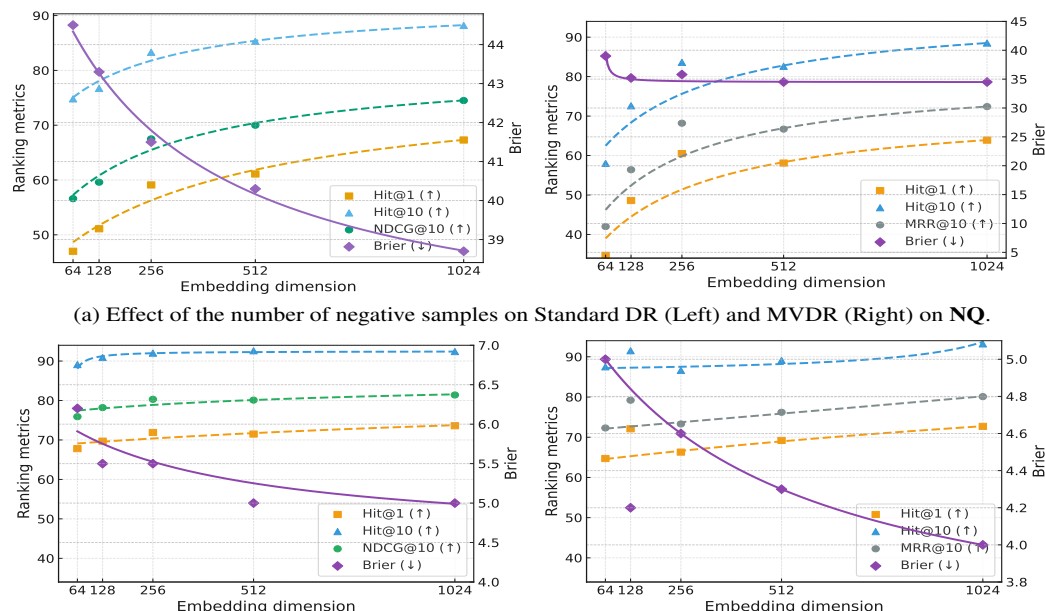

(a) Effect of the number of negative samples on Standard DR (Left) and MVDR (Right) on **NQ**.

(b) Effect of the number of negative samples on Standard DR (Left) and MVDR (Right) on **MS MARCO**.

Figure 2: DR's retrieval performance improves as the embedding dimension increases.

As shown in Table 2 (a) and Table 2 (b), both datasets exhibit the same pattern: (i) as the number of candidate documents increases, the performance of both GR and DR declines, reflecting the increased task difficulty introduced by a larger candidate pool; however, (ii) GR degrades more slowly than DR, both in magnitude and in rate. For instance, on NQ, DR's Hit@1 decreases by 6.9% and Hit@10 by 6.3%, while GR's Hit@1

Table 2: DR vs. GR under synchronized corpus scaling.

(a) On **NQ**, corpus expansion leads to a sharper degradation for DR.

| Metric | Standard DR | | | | GR-codebook | | | |
|---|---|---|---|---|---|---|---|---|
| | Initial | Final | Abs. drop | Per-unit | Initial | Final | Abs. drop | Per-unit |
| Hit@1 | 52.4 | 45.5 | 6.9 | 1.0 | 64.2 | 60.9 | 3.3 | 0.5 |
| Hit@10 | 79.9 | 73.6 | 6.3 | 0.9 | 82.5 | 79.2 | 3.3 | 0.5 |
| NDCG@10 | 61.9 | 56.5 | 5.4 | 0.8 | – | – | – | – |
| MRR@10 | – | – | – | – | 86.7 | 83.7 | 3.0 | 0.4 |

(b) On **MS MARCO**, DR likewise shows a larger performance drop than GR.

| Metric | Standard DR | | | | GR-codebook | | | |
|---|---|---|---|---|---|---|---|---|
| | Initial | Final | Abs. drop | Per-unit | Initial | Final | Abs. drop | Per-unit |
| Hit@1 | 57.5 | 48.4 | 9.1 | 1.3 | 42.3 | 39.6 | 2.7 | 0.4 |
| Hit@10 | 80.4 | 73.3 | 7.1 | 1.0 | 70.8 | 64.5 | 6.3 | 0.9 |
| NDCG@10 | 65.4 | 58.9 | 6.5 | 0.9 | – | – | – | – |
| MRR@10 | – | – | – | – | 45.1 | 41.0 | 4.1 | 0.6 |

and Hit@10 drop by only 3.3% each. This aligns with our theoretical analysis: corpus expansion amplifies the optimization drift of DR caused by local sampling, whereas GR optimizes a globally normalized objective over the full docid space for each query, making it less sensitive to additional non-relevant documents. Results for MVDR and GR-text are provided in Appendix H.

**GR and DR under model scaling.** To examine differences in model scaling, we compare GR and DR under equal added parameter budgets. We attach randomly initialized adapters of the same size to both models and train the adapters jointly with the backbone, then track ranking metrics. Note that the adapters range from 0.1B to 0.8B parameters and at the largest setting, the adapter exceeds the backbone in size, making this setup meaningful for model-scaling evaluation.

Figure 3 shows a clear upward trend for GR with the model scale. Performance improves substantially as parameters increase. On both datasets (NQ and MSMARCO), all metrics rise by roughly 5%, indicating that GR reaps sizable gains from added parameters. In contrast, DR remains flat or improves only marginally, with changes around 1%, suggesting that simply scaling parameters does not directly benefit DR. These patterns are consistent across both datasets. Taken together, the results imply that, in the era of large language models, GR is better positioned to capitalize on rapid parameter growth, whereas DR lacks an equally direct path and may require larger embeddings or richer contrastive pretraining. Please refer to Appendix I for results on MVDR and GR-text.

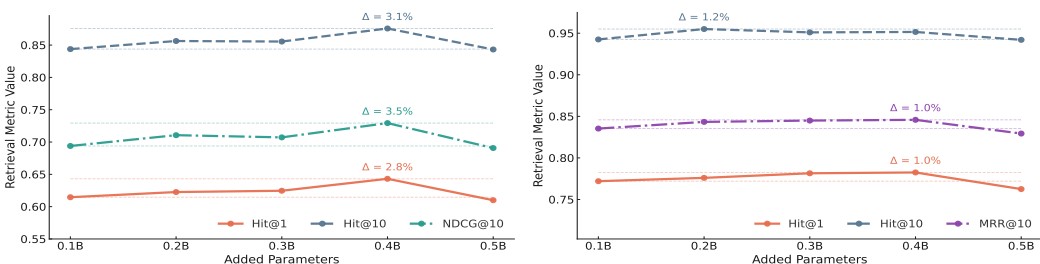

(a) Standard DR shows no clear trend of improved retrieval performance with increasing parameter scale on NQ (Left) and MS MARCO (Right).

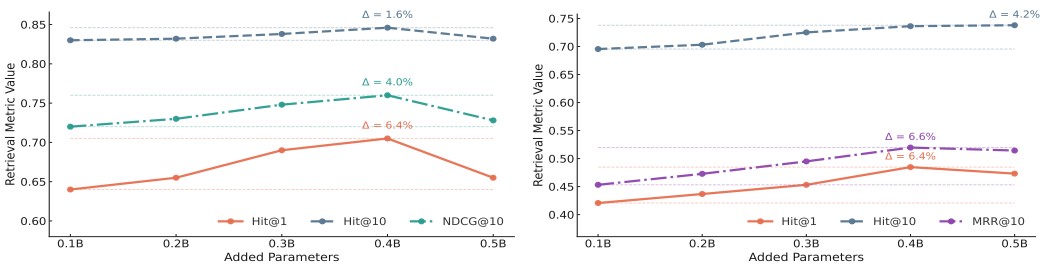

(b) GR shows a clear upward scaling trend in retrieval performance on NQ (Left) and MS MARCO (Right).

Figure 3: Comparison of DR and GR under synchronized model scaling. Only the increasing range is shown here. All models drop after 0.4B due to adding too many new parameters. See Appendix I for the full curve.

### 4.4 POTENTIAL ADVANTAGES OF GR

Next, we explore GR's advantages at larger scales using a 14B-parameter model. We focus on GR-text on the NQ dataset, as these experiments are designed to fully leverage capabilities acquired during LLM pretraining. The NQ dataset's documents are drawn from Wikipedia, with titles serving as natural text docids. Because both documents and titles are seen during pretraining, this setup directly exploits the model's world knowledge and reasoning abilities.

**Zero-shot GR.** GR performs token-by-token prediction of a docid and when the docid is textual, this inference procedure aligns with the LLM's next-token–prediction (NTP) pretraining objective. This motivates the hypothesis that an LLM can perform retrieval without any task-specific training, relying solely on its pretrained capabilities. We therefore design a zero-shot GR experiment to test this hypothesis. Specifically, we add only a prompt and enforce decoding under trie constraints, with no retrieval-specific fine-tuning.

**TTS GR.** We further assess test-time scaling (TTS) with a "think-then-retrieve" procedure to probe GR's exploitation of LLM capabilities and its internalization of the corpus. Specifically, before constrained decoding, the model first produces a short free-form reasoning snippet. The original query and the reasoning are then concatenated and passed to constrained decoding for retrieval. This augmentation is applied only at inference, while training follows the standard GR setup.

Results for zero-shot GR and TTS GR, alongside standard GR, are reported in Table 3. We summarize: (i) zero-shot GR achieves non-trivial retrieval quality (although it remains modest), suggesting that with larger models, carefully designed prompts, and suitable docids, practical training-free GR may be attainable; and (ii) even without task-specific fine-tuning, GR benefits from a pre-retrieval reasoning step, outperforming the no-reasoning baseline, which indicates that GR's parameterized internalization of documents and relevance aids retrieval via query reformulation. These experiments corroborate GR's advantages at larger model scales.

Table 3: Retrieval performance on the NQ dataset for standard GR-text and its zero-shot and TTS variants.

|  | Hit@1 | Hit@10 | NDCG@10 |
|---|---|---|---|
| Zero-shot GR | 18.1 | 23.8 | 33.3 |
| Standard GR | 45.7 | 63.5 | 88.6 |
| TTS GR | 47.3 | 65.8 | 89.1 |

## 5 Discussion

**Practical challenges of GR.** Although GR is theoretically appealing and exhibits demonstrable scaling advantages, it seldom reaches the theoretical optimum in practice, for two main reasons: (i) Noisy or biased supervision (e.g., conflicting relevance labels) and insufficient training can induce an irreducible mismatch between the learned model and the target posterior (Zhuang et al., 2022); and (ii) Prefix-constrained autoregressive decoding is prone to error propagation which means once early tokens deviate, subsequent steps tend to drift (Bevilacqua et al., 2022; Zhang et al., 2024). This issue is exacerbated when the docid design is flawed (e.g., unbalanced hierarchies, suboptimal clustering, or text-based docids that fail to cover document content). Beyond this optimality gap, engineering considerations further limit GR's practical use: (i) GR's token-by-token decoding introduces high per-step latency whereas ANN-indexed DR can provide near-instant lookups once the index is built; and (ii) under continual corpus drift, GR often needs retraining or local fine-tuning to accommodate an updated codebook or shifting hierarchical boundaries (Chen et al., 2023; Kishore et al., 2023), whereas DR commonly supports index-only updates.

**Potential solutions.** We discuss some potential solutions to address the practical challenges of GR. For *data noise and undertraining*, two complementary directions are promising: (i) treating relevance itself as the pretraining target and pretrain a decoder-only model from scratch on large-scale, noise-controlled $(q, d)$ pairs to directly optimize $-\log P(d \mid q)$, similar to some recent works on generative recommendation (e.g., one-rec (Deng et al., 2025)). This is appropriate when relevance is explicitly defined by human rules (e.g., e-commerce query–item (Rajput et al., 2023), ads matching (Fan et al., 2019), FAQ–KB pairs (Sakata et al., 2019)); and (ii) exploiting the world knowledge and reasoning of LLM bases. Specifically, teach the model the semantics and interface of retrieval with light instruction tuning instead of memorizing full-corpus relevance. At inference, execute "retrieval as constrained generation" via constrained decoding. This is suitable when the relevance underlying the retrieval task is already encoded in the pretraining corpus (e.g., Wikipedia or encyclopedic retrieval (Petroni et al., 2020)).

For *early-error propagation*, relaxing clustering constraints or decoding constraints might work. Specifically, allowing each document to belong to multiple clusters (especially for boundary cases) might reduce early-errors. On the decoding side, enabling backoff mechanisms or, when necessary, allowing tokens outside the constraint set to recover from early mistakes.

For *engineering efficiency*, integrating GR with DR in a single system within a single system is promising. One practical design is to let GR decode only a shallow prefix to perform coarse-grained category recall, followed by DR for fine-grained retrieval within that category. This coarse-to-fine design is expected to leverages GR's capacity to fit relevance while mitigating error accumulation and reducing the latency associated with deep prefix-constrained decoding down to docids.

## 6 Conclusion and limitations

We have systematically compared DR and GR in terms of learning objectives and representational capacity. Theoretically, GR performs globally normalized maximum likelihood over the docid space, thereby avoiding the calibration gap introduced by DR's locally normalized contrastive learning. Moreover, under fixed bilinear interactions, DR is constrained by a low-rank bottleneck determined by the embedding dimension, whereas GR admits higher-rank approximations. Empirically, results on the NQ and MS MARCO datasets show that calibration and ranking metrics corroborate these theoretical differences. Under comparable corpus and parameter scaling, GR achieves larger gains and further demonstrates advantages in zero-shot and test-time scaling. In summary, GR shows promise in overcoming DR's bottlenecks, though several practical challenges remain.

This work also has several limitations: (i) our theoretical analysis assumes idealized formulations of GR and DR and does not fully account for the effects of training data, docid design, or decoding/search strategies; (ii) due to resource constraints, we were unable to compare GR and DR at larger model and corpus scales; (iii) our comparisons did not include state-of-the-art variants of GR and DR; and (iv) although we propose several potential extensions for GR, we did not conduct preliminary experiments to validate their effectiveness.

## 7 REPRODUCIBILITY STATEMENT

We summarize the steps we have taken to ensure reproducibility and point to where the relevant details can be found. The theoretical assumptions are stated in Section §3 and Appendices A–E, where we provide complete proofs for the CE–KL decomposition, the DR local-normalization gap, the low-rank bottleneck, and the universality of GR. Readers can map each claim in Section §3 to its corresponding appendix proof. Our experimental setup, including model choices, datasets, evaluation metrics, and training/inference details are specified in Section §4.1 and Appendix F. We enumerate all experimental factors that affect the results (the size/quality of negative samples, embedding dimensionality, and corpus/model scaling) and provide their implementations and settings in Appendix G. For large-model experiments (zero-shot GR and TTS GR), we report implementation details in Appendix G including the exact instructions/prompts.

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

# A   CROSS-ENTROPY AND KL DECOMPOSITION

For completeness, we give a concise derivation of Eq. 6. Let $P$ be the data distribution and $Q_\Theta$ the model on the same finite support. By definition,

$$\mathrm{CE}(P, Q_\Theta) = \mathbb{E}_{x \sim P}\big[ -\log Q_\Theta(x)\big] = \mathbb{E}_{x \sim P}\Big[ \log \frac{P(x)}{Q_\Theta(x)}\Big] + \mathbb{E}_{x \sim P}\big[ -\log P(x)\big]. \qquad (7)$$

The first term equals $\mathrm{KL}(P\|Q_\Theta)$ and the second equals $H(P)$, hence $\mathrm{CE}(P, Q_\Theta) = H(P) + \mathrm{KL}(P\|Q_\Theta)$. For conditional sequence models (GR), summing token-wise cross-entropies yields the same identity after taking expectations over queries.

# B   PROOF OF THEOREM 3.1

For a query $q$, define the global and in-batch partition functions

$$Z(q) = \sum_{d' \in \mathcal{D}} \exp\big(S(q, d')/\tau\big), \qquad Z_K(q) = \sum_{d' \in \{d^+\} \cup \mathcal{N}(q)} \exp\big(S(q, d')/\tau\big). \qquad (8)$$

Then

$$\log \widetilde{P}_\Theta(d^+ \mid q) - \log P_\Theta(d^+ \mid q; \mathcal{N}) = \log Z_K(q) - \log Z(q). \qquad (9)$$

Let $\mu$ be the corpus marginal (uniform over $\mathcal{D}$) and $\pi$ the negative-sampling proposal,

$$\delta(q) = \log \mathbb{E}_{d \sim \pi}\big[\mathrm{e}^{S(q,d)/\tau}\big] - \log \mathbb{E}_{d \sim \mu}\big[\mathrm{e}^{S(q,d)/\tau}\big]. \qquad (10)$$

Taking expectation over the sampling of $\mathcal{N}(q)$ and using Jensen's inequality,

$$\mathbb{E}\big[\log Z_K(q)\big] \geq \log \mathbb{E}\big[Z_K(q)\big] \geq \log K + \log \mathbb{E}_{d \sim \pi}\big[\mathrm{e}^{S(q,d)/\tau}\big], \qquad (11)$$

where we use the fact that $\mathbb{E}[Z_K(q)] \geq K \mathbb{E}_{d \sim \pi}[\mathrm{e}^{S(q,d)/\tau}]$. Since $Z(q) = N \mathbb{E}_{d \sim \mu}[\mathrm{e}^{S(q,d)/\tau}]$, we obtain

$$\mathbb{E}\big[\log Z_K(q) - \log Z(q)\big] \geq \log \frac{K}{N} - \delta(q). \qquad (12)$$

Averaging over queries gives Theorem 3.1.

# C   CONSTRUCTIVE UNIVERSALITY FOR GR

Fix a bijection between $\mathcal{D}$ and the leaves of a $|\mathcal{V}|$-ary trie of depth $L$. Given a target posterior $P^\star(\cdot \mid q)$, assign at each internal node the conditional distribution over its children to match the subtree mass under $P^\star$: for node $u$ with children $\{v\}$, set

$$p^\star(v \mid u, q) = \frac{\sum_{\text{leaves } \ell \in \mathrm{subtree}(v)} P^\star(\ell \mid q)}{\sum_{\text{leaves } \ell \in \mathrm{subtree}(u)} P^\star(\ell \mid q)}. \qquad (13)$$

A decoder with sufficient capacity can approximate each local conditional $p^\star(\cdot \mid u, q)$ arbitrarily well. By the chain rule along any root-to-leaf path, the product of these conditionals approximates the target leaf mass, hence the induced leaf distribution approaches $P^\star(\cdot \mid q)$ in total variation. Under prefix-constrained decoding, the same construction applies because valid leaves are exactly the trie leaves corresponding to $\mathcal{D}$.

# D   LOW-RANK LIMITATION FOR DR

Let $S^* \in \mathbb{R}^{m \times N}$ be a ground-truth logit matrix whose $(i, j)$-entry is a monotone transform of $\log P^*(d_j \mid q_i)$. Any bilinear DR model with embedding dimension $r$ factorizes as $S = QD^\top$ and thus $\mathrm{rank}(S) \leq r$ (or $\leq cr$ with $c$ independent interaction channels). By the Eckart-Young-Mirsky theorem,

$$\min_{\mathrm{rank}(S) \leq r} \|S - S^*\|_F^2 = \sum_{i > r} \sigma_i(S^*)^2, \qquad (14)$$

the squared Frobenius norm of the spectral tail beyond rank $r$.

Consequently, if $S^*$ has a heavy spectral tail, any fixed-$r$ DR model incurs an irreducible posterior approximation error unless $r$ (or the number of interaction channels) is increased.

# E  A HIGH-PROBABILITY BOUND FOR $\log Z_K - \log Z$

Fix a query $q$ and define $X = e^{S(q,d)/\tau}$ for $d \sim \pi(\cdot \mid q)$ with mean $\mu_\pi = \mathbb{E}_\pi[X]$ and variance $\sigma_\pi^2 = \mathrm{Var}_\pi[X]$. Let $X_1, \dots, X_K$ be i.i.d. copies and $\bar{X}_K = \frac{1}{K}\sum_{i=1}^{K} X_i$. Assuming $X$ is sub-exponential (e.g., bounded or with a finite moment generating function in a neighborhood of 0), a Bernstein-type inequality gives, for any $\epsilon \in (0,1)$,

$$\Pr\left[\ \log \bar{X}_K \ \leq \ \log \mu_\pi \ - \ \epsilon\ \right] \ \leq \ \exp\left(-\ \frac{K\,\epsilon^2}{2(\sigma_\pi^2/\mu_\pi^2 + \epsilon/3)}\right). \tag{15}$$

Since $Z_K(q) = \sum_{d \in \mathcal{N}(q)} e^{S(q,d)/\tau} = K\,\bar{X}_K$ and $Z(q) = N\,\mu_\mu$ with $\mu_\mu = \mathbb{E}_{d\sim\mu}[e^{S(q,d)/\tau}]$, we have with probability at least $1 - \exp(-cK\epsilon^2)$ (for a constant $c$ depending on moments of $X$):

$$\log Z_K(q) - \log Z(q) \ \geq \ \log \frac{K}{N} \ - \ \left(\log \mu_\mu - \log \mu_\pi\right) \ - \ \epsilon \ = \ \log \frac{K}{N} \ - \ \delta(q) \ - \ \epsilon. \tag{16}$$

Averaging over $q$ yields a high-probability version of Theorem 3.1. We emphasize that this bound holds under i.i.d. negatives from $\pi$; for adaptive or "hard-negative" proposals $\pi_t(\cdot \mid q, \Theta_t)$, the same form holds with an additional bias term in $\delta_t(q)$ that captures proposal/model dependence.

# F  DETAILED EXPERIMENTAL SETUP

**Datasets.** We evaluate on two standard retrieval benchmark datasets: (i) **Natural Questions (NQ)** (Kwiatkowski et al., 2019). This is a collection of real-user questions paired with supporting Wikipedia evidence. We use the official train (313K) and test (7K) splits. To make generative retrieval feasible, we ensure that each test query's gold document appears in the docid inventory constructed from the training corpus (i.e., the gold docid is seen during training); and (ii) **MS MARCO Passage** (Bajaj et al., 2016). This is a set of web search queries from Bing with associated passages. We use the passage-ranking subset and sample 1M training pairs and 2K evaluation queries from the official train/test splits. Unlike NQ, we do not enforce the "seen-document" constraint on MS MARCO (because enforcing it would shrink the evaluation set to only few hundred queries).

**Models used for comparison.** We implement two representative systems for both DR and GR and intentionally avoid complex variants to keep comparisons fair and transparent. For DR, we implement (i) a *standard bi-encoder* in the spirit of DPR (Karpukhin et al., 2020) with inner-product scoring; and (ii) a *multi-vector late-interaction* variant like ColBERT v1 (Khattab & Zaharia, 2020). For GR, we implement two varying about the docid design and train/inference follow the DSI-style (Tay et al., 2022): (i) *codebook docids* built via residual quantization, each docid is a length-6 sequence of 8-bit code indices; and (ii) *textual docids* that directly use the title as the document identifier. All GR decoding is prefix-constrained by a trie constructed from the set of valid docids.

**Metrics.** We report the calibration metric *Brier*, which is the mean squared error between the predicted relevance probability and the ground truth over the query's rank-1 candidate. We report unnormalized (raw) Brier scores, consequently, they are comparable only within the same dataset and experimental series, and the values are not comparable across experiments. We also report four retrieval metrics: (i) *Hits@k* indicates whether at least one relevant document appears in the top-$k$ results for a query; (ii) *NDCG@k* is the normalized discounted cumulative gain at cutoff $k$, using binary gains with logarithmic discounting by rank; and (iii) *MRR@k* is the mean reciprocal rank of the first relevant document within the top-$k$.

**Training and inference.** To control for capacity and pretraining, all DR models are built on Qwen3-Embedding-0.6B, and all GR models use Qwen3-0.6B (Yang et al., 2025a). Unless otherwise noted, we train with the Adam optimizer (Kingma & Ba, 2014) using its default settings. At inference time, DR retrieves top-$k$ candidates using FAISS-based ANN search (Xiong et al., 2020b), while GR performs top-$k$ constrained decoding over the docid trie.

# G  DETAILED EXPERIMENTAL IMPLEMENTATION

**DR negative sampling.** The goal is to assess how negative sampling affects DR performance along two dimensions: size and quality. For size, we use random negatives and vary the number of negatives during training. For quality, we experiment only on NQ, which provides both standard and

hard negatives: we vary the proportion of hard negatives in the sampled batch. If the official hard negatives are insufficient, we first fill with the provided standard negatives, and if still insufficient we complete the batch with random negatives. In this experiment, both query and document embedding dimensionality is fixed at 128, and MVDR and DR share identical settings.

**DR embedding size.** The goal is to examine the constraint imposed by the embedding dimension on DR. We append a two-layer non-linear projection (ReLU activations) after the model's output layer to map embeddings to the target dimension and this projection is trained jointly with the backbone. Random negative sampling is used, and MVDR shares the same settings as DR.

**Corpus scaling.** The goal is to observe how GR and DR behave when the training corpus size is increased by the same amount. We control the number of documents in the corpus and require that each document appears at least once as a positive in the training set; the test set is a subset of this corpus. In this experiment, DR uses random negative sampling and 128-dimensional embeddings. GR-codebook and GR-text follow the configurations described in the main text. GR-text is evaluated only on NQ, where the official titles can serve as textual docids.

**Model scaling.** The goal is to compare GR and DR when model capacity is scaled by the same budget. We equip each layer with randomly initialized adapters of matched size and control the scaling by the total number of newly introduced parameters and adapters are trained jointly with the backbone. Note that the largest adapter budget can exceed the original backbone size. All other settings mirror those in the Corpus Scaling experiment.

**GR zero-shot.** The goal is to evaluate GR's retrieval ability without fine-tuning, relying solely on pretrained knowledge. This experiment is conducted only on NQ with the GR-text, because NQ's documents and their titles (used as docids) come from Wikipedia which is thoroughly covered during LLM pretraining making zero-shot GR feasible. We employ a larger model (Qwen3-14B) for this study. Specifically, we do not fine-tune Qwen3-14B, instead, we prepend a prompt to each query: `Given the question, predict the document title that most likely contains the answer. The title is:` and then enforce trie-constrained decoding to produce the docid.

**GR TTS.** The goal is to assess whether GR can leverage an LLM's reasoning capability and its internalized document knowledge to improve performance via a "think-then-retrieve" procedure. This experiment is conducted only on NQ with the GR-text, using Qwen3-14B as the backbone. During training, We prepend a retrieval instruction $I_r$ to each query: `Given the question, predict the document title that most likely contains the answer. The title is:` and fine-tune GR with LoRA. During inference, the model first performs unconstrained "thinking" given the prompt: `Briefly think about the document title that may contain the answer to this question.` The generated reasoning is then concatenated with the original query and the retrieval instruction $I_r$, and constrained decoding is applied to produce the docid.

# H EXTENDED RESULTS OF CORPUS SCALING

This section supplements the corpus scaling experiments in Section 4.3. Figure 4 presents the full performance trends under corpus scaling for all models (including MVDR and GR-text, which are not covered in the main text Table 2). The conclusions mirror those in the main text: overall, DR exhibits a larger performance drop than GR as the corpus size increases.

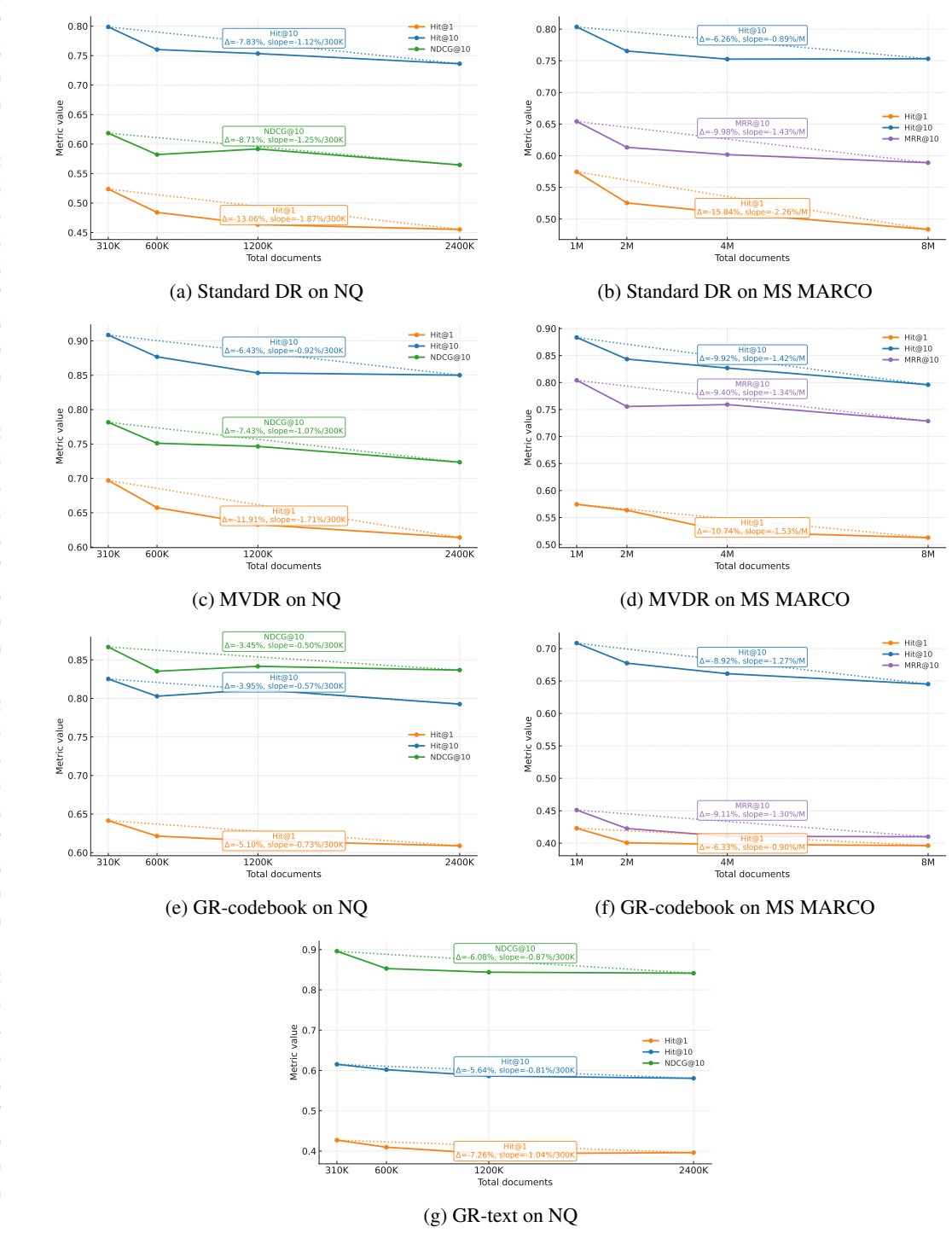

Figure 4: Extended results of corpus scaling.

# I    EXTENDED RESULTS ON MODEL SCALING

This section supplements the model scaling experiments in Section 4.3. Figure 5 presents the full performance trends under model scaling for all models (including MVDR and GR-text, which are not covered in the main text, Figure 3). The end-of-curve downturn observed in all traces is likely due to the addition of excessive parameters. Ignoring this effect and focusing on the initial stage where model scaling yields gains, the conclusion aligns with the main text: GR derives greater benefits from increases in parameter scale.

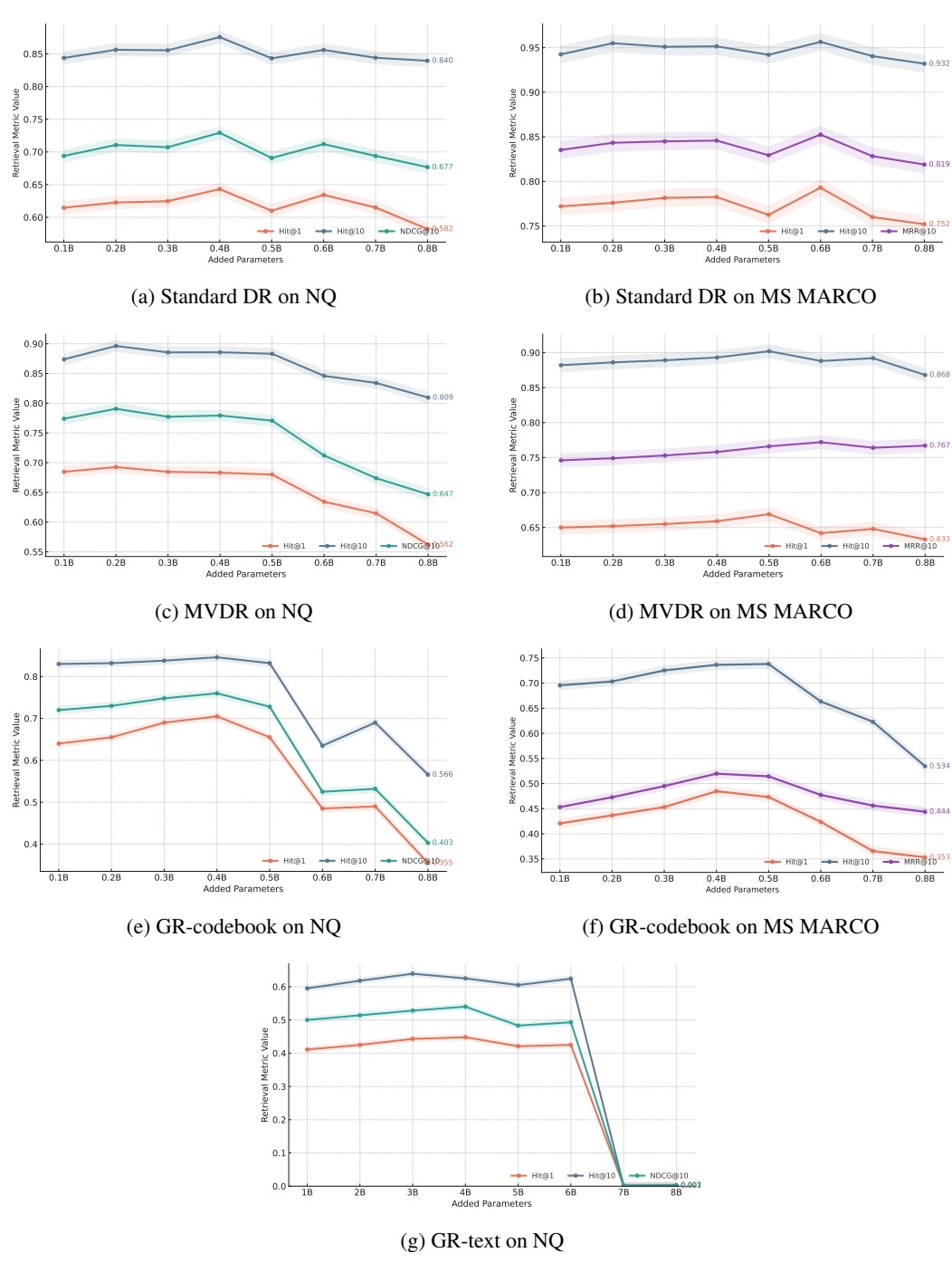

Figure 5: Extended results of model scaling.

## J    LLM USAGE

We used a large language model to help polish wording and to generate codes for data visualization. All core ideas, theoretical analysis, experimental design, and the initial full manuscript were conceived and written by all co-authors. All LLM-suggested text and code were reviewed and verified by the authors before inclusion.

