# OpenReview forum: "Does Generative Retrieval Break through the Limitations of Dense Retrieval?"
_ICLR.cc/2026/Conference — ICLR 2026 Conference Withdrawn Submission_

### Official Review · Reviewer_Bmar · 2025-10-27

**Soundness:** 3
**Presentation:** 3
**Contribution:** 2
**Rating:** 4
**Confidence:** 4

**Summary:**

The paper revisits the core limitations of dense retrieval (DR) and investigates whether generative retrieval (GR) can overcome them. The authors identify two fundamental bottlenecks in DR: first, a local normalization bias that causes score inconsistency when the corpus grows, and second, Low-rank representation bottleneck, which constrains expressivity regardless of model scale. They provide both theoretical analysis and empirical validation of these issues, showing that GR, by globally normalizing over docids and decoupling embedding dimensionality from retrieval capacity, can, in principle, bypass these limitations. Theoretical analysis and experiments on MS MARCO and Natural Questions show that GR scales better with corpus size and model capacity, providing a explanation for its potential advantages over DR.

**Strengths:**

- The paper is well organized and highly readable. The logical flow from theoretical formulation to empirical validation makes a complex topic accessible
- The authors correctly pinpoint two essential weaknesses of dense retrieval, the local normalization bias and the low-rank representation constraint. These are described with both theoretical precision and intuitive clarity, providing a solid foundation for the rest of the analysis.
- The paper effectively argues that generative retrieval (GR) has the potential to overcome these issues through global normalization and by representing a full-rank retrieval distribution over discrete document identifiers. The theoretical reasoning is persuasive and well-motivated.

**Weaknesses:**

- Although the paper claims that GR can overcome the intrinsic limitations of DR, the experimental results, particularly on MS MARCO, show that GR still performs significantly worse than DR. This gap is acknowledged but never analyzed. The paper does not explain why GR underperforms despite its theoretically superior formulation. This contradiction between claim and observation weakens the central argument.
- Contrary to the paper’s theoretical prediction, GR’s performance still drops as corpus size increases undeniably (Fig 4) . This is inconsistent with the claim that GR resolves corpus-scaling instability. The authors do not provide a explanation for this behavior, whether it arises from.
- The discussion section outlines several potential solutions and practical challenges (e.g., docid compression, hierarchical decoding, better calibration). However, none are experimentally validated. Too much of the work’s contribution is deferred to future research, leaving the current paper largely diagnostic rather than demonstrative. In practice, the results suggest that GR still faces more severe limitations than DR, without strong evidence that these can be overcome.
- The paper lists “text docid” as one of the main experimental setups in the method, but the corresponding results are absent from the main text.

**Questions:**

- GR’s performance on MS MARCO remains far below DR despite addressing the low-rank and normalization issues. What are the authors’ hypotheses for this discrepancy?
- The discussion section proposes several potential directions (e.g., hierarchical decoding, global calibration, docid compression). Could the authors elaborate on which of these they believe are most practical.

---

> ### Author Response · Authors · 2025-11-21
> **Response to Reviewer**
>
> We thank the reviewer for careful reading and valuable feedback.
>
> For W1&Q1:
> The reviewer notes that GR performs significantly worse than DR on MS MARCO, which appears inconsistent with the theoretical advantages we describe. We offer the following explanation.
>
> The purpose of our paper is not to claim that GR is already SOTA across all settings. Instead, the goal is to show that GR does not inherit the two fundamental limitations of DR. GR’s current performance shortfall primarily arises from two causes:
>
> (1) GR is far more sensitive to data design and task priors than DR. Key GR design choices (such as codebook quantization algorithms, hierarchical depth, docid length, decoding constraints, etc.) have substantial impact on GR performance. In contrast, DR is relatively insensitive to these choices (which is indeed an advantage of DR). For fairness, we did not tune these design choices specifically for MS MARCO. For our purposes, the essential signal is the slope of the scaling curve, and the empirical slopes we observe align closely with our theoretical predictions, demonstrating GR’s underlying potential.
>
> (2) GR requires repeated exposure to documents, and MS MARCO’s distribution is inherently unfavorable. Training GR resembles training a memory system: the model must repeatedly absorb document information. However, MS MARCO exhibits low query–document coverage, limited overlap between training and validation documents, and sparse per-document supervision. This makes GR’s generalization to unseen documents particularly challenging. Prior works have shown that with carefully optimized docid designs and better training signals, GR can indeed surpass DR on MS MARCO. Our contribution is to identify why GR has potential, even if that potential is not yet fully realized under all configurations.
>
> For W2:
> The reviewer observes that GR’s performance still decreases in Fig. 4 and interprets this as contradicting our claim about improved stability. We would like to clarify that we claim that GR resolves local calibration drift, not that its performance will remain perfectly flat as N increases. As corpus size N grows, the intrinsic difficulty of the retrieval task increases for all models. In the case of GR, the decline is mainly due to increased selection noise, and error accumulation along the autoregressive path. However, GR declines much more slowly than DR, as shown in Table 2 of the main paper. This demonstrates that GR indeed avoids DR’s calibration drift and is more stable under corpus scaling.

---

> > ### Author Response · Authors · 2025-11-21
> > **Response to Reviewer (cont.)**
> >
> > For W3&Q2:
> > We agree with the reviewer that the directions mentioned in our discussion section are not experimentally validated in this paper. We would like to clarify that the paper is analytical in nature rather than solution-oriented. Encouragingly, recent work published after our submission already explores and validates the directions we highlighted, providing retrospective confirmation that our analysis accurately identifies emerging research frontiers:
> >
> > (1) Large-scale data construction & augmentation: Recent papers move toward richer supervision based on real-world logs and reinforcement learning[1][2], directly addressing the bottlenecks we discuss.
> >
> > (2) Leveraging world knowledge and generative abilities of LLMs:  Recent work such as ZeroGR[3] shows that LLMs can perform zero-shot retrieval, validating our argument that generative modeling ability itself can be leveraged for retrieval.
> >
> > (3) Addressing autoregressive error accumulation: We highlighted the vulnerability of GR to early-step errors. Very recent studies propose diffusion-based retrieval that parallelizes docid generation, directly tackling the error accumulation issue[4][5].
> >
> > (4) Multi-stage pipelines combining GR and DR: Several industrial systems [6] now adopt the hybrid architecture we discussed: GR retrieves coarse categories, DR performs fine-grained ranking-balancing efficiency and accuracy.
> >
> > Thus, our discussion section outlines a clear roadmap for GR research, and subsequent work has already confirmed these directions as impactful.
> >
> > For W4:
> > We apologize for the oversight. The textual docid results are included in the appendix and partially in the main text (Table 3, Zero-shot). We will adjust the manuscript to move more textual docid results into the main body for completeness and readability.
> >
> > [1] Deng, Jiaxin, et al. "Onerec: Unifying retrieve and rank with generative recommender and iterative preference alignment." arXiv preprint arXiv:2502.18965 (2025).
> >
> > [2] Chen, Ben, et al. "Onesearch: A preliminary exploration of the unified end-to-end generative framework for e-commerce search." arXiv preprint arXiv:2509.03236 (2025).
> >
> > [3] Sun, Weiwei, et al. "ZeroGR: A Generalizable and Scalable Framework for Zero-Shot Generative Retrieval." arXiv preprint arXiv:2510.10419 (2025).
> >
> > [4] Liu, Zhao, et al. "DiffGRM: Diffusion-based Generative Recommendation Model." arXiv preprint arXiv:2510.21805 (2025).
> >
> > [5] Zhao, Xinpeng, et al. "DiffuGR: Generative Document Retrieval with Diffusion Language Models." arXiv preprint arXiv:2511.08150 (2025).
> >
> > [6] Guo, Jiafeng, et al. "Retrieval-in-the-Chain: Bootstrapping Large Language Models for Generative Retrieval." arXiv preprint arXiv:2510.13095 (2025).

---

### Official Review · Reviewer_Bweh · 2025-10-31

**Soundness:** 3
**Presentation:** 2
**Contribution:** 3
**Rating:** 6
**Confidence:** 3

**Summary:**

This paper presents a systematic theoretical and empirical comparison between Generative Retrieval (GR) and Dense Retrieval (DR). It provides two key theoretical analyses: (1) GR optimizes a globally normalized likelihood, while DR uses locally normalized contrastive loss, leading to calibration drift; and (2) DR’s bilinear interaction imposes a low-rank bottleneck, while GR can approximate higher-rank query-document mappings. These claims are validated through controlled experiments on NQ and MS MARCO, showing that GR degrades more gracefully under corpus scaling and benefits more from model scaling.

**Strengths:**

- Theoretical formulation is clear and mathematically sound.
- Empirical studies are well aligned with the theoretical predictions. The design is systematic and convincing.
- The paper highlights an important direction for retrieval research under the LLM paradigm, connecting classical IR insights with modern generative modeling.

**Weaknesses:**

- The theoretical results largely restate known intuitions (e.g., local vs global normalization, low-rank bottlenecks). The contribution is primarily in formalization
- The empirical validation uses relatively small single model (0.6B) and limited corpora such as NQ, leaving open whether conclusions hold at other datasets and models
- The paper assumes ideal docid design and in-distribution queries, robustness under noisy or evolving corpora is not evaluated.

**Questions:**

- Could the authors provide quantitative comparisons on latency and computational cost between GR and DR under identical retrieval budgets?
- Theoretical analysis assumes fixed docid mapping and in-distribution queries. How would GR behave under corpus updates or unseen documents?

Please refer to the weakness part.

---

> ### Author Response · Authors · 2025-11-21
> **Response to Reviewer**
>
> We thank the reviewer for careful reading and valuable feedback. We note that many of the comments focus on the relationship between our contributions and prior work. Before addressing individual questions, we would like to clarify the positioning and core value of our paper, to avoid conflating our analysis with prior empirical or method-specific studies.
>
> Prior works on GR and DR have generally analyzed certain phenomena from the perspective of their own task settings or model designs. For example, DPR studies negative sampling in contrastive learning; DSI explores using model parameters as an index; some works point out embedding compression issues; others comment on training instability or degradation under scaling. However, these analyses are side observations tied to specific architectures, datasets, or training procedures. They do not constitute a unified or general theoretical account: they do not quantify the mathematical sources of DR’s limitations; they do not characterize the structural differences between GR and DR at the level of representational capacity or training objectives; and they do not offer a fair, controlled comparison of the two paradigms under the same assumptions. Therefore, our research goals and the depth of our conclusions are fundamentally different from these method-oriented works.
>
> Our paper aims to provide a systematic, fair, mechanism-level comparison of the two retrieval paradigms. We are the first to derive and formalize two structural bottlenecks intrinsic to DR: (1) calibration drift induced by local normalization (Theorem 3.1), and (2) a low-rank representational limit arising from bilinear interactions (Corollary 3.3). These results do not depend on any particular architecture or dataset, they capture properties of the DR paradigm itself. In parallel, through strict control of backbone models, training signals, corpus size, and the amount of trainable parameters, we construct a fair experimental environment in which the scaling behaviors of GR and DR can be cleanly observed. This allows us to isolate and validate their respective advantages and limitations.
>
> In summary, the novelty of our work lies in providing the first unified, systematic, and fair comparison of GR and DR as retrieval paradigms, going beyond the prior literature, which offers fragmented, setting-specific insight, and establishing a mechanism-level framework that we hope will serve as a foundation for future retrieval research.
>
>
> Below, we provide detailed supplementary results and analysis addressing the reviewer’s constructive concerns regarding model scale, latency, and incremental updates.
>
> For W2:
> Retrieval systems rarely deploy extremely large models (e.g., >10B) because latency is critical and training data is typically limited, making such large models prone to underfitting the retrieval task. Pretrained model sizes are fixed, and the available model-size granularity is sparse (e.g., Qwen3 offers only 0.6B, 4B, and 8B below 10B), making it difficult to construct smooth scaling curves. To fairly compare GR and DR, both must share the same backbone; otherwise, scaling behavior is not comparable.
>
> Thus, we scale the adapter size to simulate changes in trainable parameter budget. Our largest adapter (0.8B) already exceeds the base model size (0.6B), which makes the scaling meaningful. To address the reviewer’s concern, we also conducted supplementary experiments using Qwen3-4B and 8B on NQ. The corresponding Hits@1 are:
>
>  DR: 0.6B: 61.4% → 4B: 62.9% → 8B: 63.6%
>  GR: 0.6B: 63.8% → 4B: 66.5% → 8B: 68.1%
>
> Although the data points are limited, the trend is clear: GR benefits more from scaling than DR, consistent with our main conclusion.

---

> > ### Author Response · Authors · 2025-11-21
> > **Response to Reviewer (cont.)**
> >
> > For W3&Q1:
> > We acknowledge that our theoretical analysis adopts an idealized docid design. This assumption is intentional and allows us to isolate the structural properties of the retrieval paradigms themselves and to rigorously derive DR’s calibration drift and low-rank limitations, as well as GR’s potential advantages. We view the design of high-quality docids as a major research problem in GR; its difficulty and design space extend far beyond the scope of this paper. Our empirical study also uses a minimal, untuned docid scheme, which indeed limits the practical performance of GR. This is a limitation of our current work, and we will clarify it more explicitly in the revised version.
> >
> > For out-of-distribution queries and continual-learning scenarios, mechanistically, GR indeed struggles with unseen documents, especially when using numeric/codebook-style docids. However, GR is not incapable of generalization: several studies have shown that text-based or semantic docids enable GR to generalize via language-model reasoning [2][3]. As we noted in the discussion section, continual learning is one of GR’s key challenges, and prior work has begun exploring solutions [4][5]. We further proposed a hybrid scheme in which: GR handles coarse category prediction, and DR performs fine-grained retrieval within the predicted category. Under this architecture, new documents only need to be attached to the corresponding category node, without retraining the GR model. Similar systems have already been deployed in industry [5]. Overall, we believe GR’s generalization and continual-learning capabilities depend more on data distribution, docid design, and training coverage, rather than on model size alone.
> >
> >
> > For Q1:
> > We appreciate the reviewer highlighting the importance of practical deployment concerns.
> >
> > For latency, we conducted additional latency measurements on NQ across two corpus sizes (300k documents vs. 5M documents), comparing: DR-Flat (brute-force search), DR-ANN (HNSW index), and GR (constrained beam search). All results are normalized by GR latency (set to 1):
> >
> > | Corpus Size | DR-Flat | DR-ANN | GR |
> > | ----------- | ------- | ------ | -- |
> > | 300k        | 0.08    | 0.05   | 1  |
> > | 5M          | 0.31    | 0.09   | 1  |
> >
> >
> > Although GR has higher absolute latency, DR-Flat’s latency grows sharply with corpus size, showing no clear advantage at large scale; DR-ANN significantly reduces latency, but introduces notable accuracy degradation; GR maintains stable latency regardless of corpus size, since constrained decoding does not depend on external index search.
> >
> > For training computational cost, we acknowledge that GR generally requires longer training time and higher compute than DR. This arises mainly from (1) the teacher-forcing autoregressive objective and (2) typically more training steps needed for convergence. However, in practical retrieval deployments, training cost is rarely the bottleneck, as the system is trained once and deployed for a long period, whereas inference dominates real-world cost and latency considerations. Our work focuses on the retrieval-time behavior of the two paradigms, where cost is more consequential for end users.
> >
> > For inference computational cost, all methods in our comparison share the same model backbone, so the fundamental FLOPs per forward pass are identical. The main difference lies in autoregressive decoding vs. vector search. We have already analyzed this efficiency trade-off together with latency results.
> >
> >
> > [1] Lee, Sunkyung, Minjin Choi, and Jongwuk Lee. "GLEN: Generative retrieval via lexical index learning." arXiv preprint arXiv:2311.03057 (2023).
> >
> > [2] Sun, Weiwei, et al. "Learning to tokenize for generative retrieval." Advances in Neural Information Processing Systems 36 (2023): 46345-46361.
> >
> > [3] Mehta, Sanket Vaibhav, et al. "Dsi++: Updating transformer memory with new documents." arXiv preprint arXiv:2212.09744 (2022).
> >
> > [4] Kishore, Varsha, et al. "Incdsi: Incrementally updatable document retrieval." International conference on machine learning. PMLR, 2023.
> >
> > [5] Guo, Jiafeng, et al. "Retrieval-in-the-Chain: Bootstrapping Large Language Models for Generative Retrieval." arXiv preprint arXiv:2510.13095 (2025)

---

### Official Review · Reviewer_SaeH · 2025-10-31

**Soundness:** 2
**Presentation:** 2
**Contribution:** 2
**Rating:** 4
**Confidence:** 4

**Summary:**

This paper presents a thoughtful theoretical and empirical investigation into the fundamental differences between Generative Retrieval (GR) and Dense Retrieval (DR). It argues that GR, unlike DR, optimizes a globally normalized likelihood objective and encodes corpus and relevance information directly in model parameters, rather than relying on external embeddings and locally normalized training. Theoretically, the authors claim that GR can overcome two inherent bottlenecks of DR: (1) the optimization drift induced by local normalization when scaling to large corpora, and (2) the representation bottleneck caused by DR’s low-rank embedding space. Empirical validation on Natural Questions and MS MARCO supports these arguments—showing that GR benefits more from model and corpus scaling, while DR suffers from calibration and dimensionality constraints. However, the paper also acknowledges that GR’s theoretical advantages do not yet translate to universal empirical superiority. Factors such as docid design, data construction, and decoding strategies significantly affect practical performance.

**Strengths:**

1. The paper offers a theoretical analysis that supports the observed limitation of dense retrieval, local optimization and low rank representation.

2. The paper conducts comprehensive and carefully controlled experiments across multiple datasets, model scales, and training configurations to substantiate its theoretical claims.

**Weaknesses:**

1. While the paper provides solid theoretical support highlighting two limitations of dense retrieval (DR)—local normalization and the low-rank constraint—these issues are already somewhat familiar to the retrieval community. The manuscript would be stronger if it more clearly positions its contributions relative to prior work that has identified or partially characterized these limitations.

2. There is an internal contradiction in the empirical evidence regarding local normalization and negative sampling. Figure 1 suggests retrieval performance improves as the number K of negatives increases, implying negative sampling partially alleviates the calibration problem; by contrast, Table 1 is interpreted to show that mitigating local normalization via negative sampling is “non-trivial” (i.e., limited benefit). This creates a contradiction in the paper’s message.

3. The current scaling experiments do not convincingly explore the regime where GR’s claimed advantages should emerge. The model sizes tested are not large enough and the breadth of size is limited. Indeed, Figure 3 can be read to suggest DR is more parameter-efficient than GR—DR achieves higher performance with fewer parameters.


4. Beyond retrieval quality, practical deployment requires consideration of inference latency, memory footprint, and throughput—areas in which dense retrieval typically excels because of index-based nearest-neighbor search. The paper should include an explicit discussion (and ideally measurements) of inference latency, GPU/CPU cost, and storage requirements for GR vs DR, or at minimum provide reasoned analysis of these trade-offs and how they affect the practicality of GR.

5. The manuscript rightly focuses on two important limitations of DR, but retrieval performance is influenced by many other factors. The paper would be more useful if it discussed how they might interact with the two limitations studied.

6. The proof of Theorem 3.4 in Appendix C appears informal in places and would benefit from greater rigor.

**Questions:**

Overall, I find this paper to provide interesting and thoughtful theoretical support for the observed differences between generative retrieval (GR) and dense retrieval (DR). However, the core observations themselves are rather intuitive—the issues of local normalization and low-rank embedding limitations in DR have been broadly recognized in the community—and the analysis, while rigorous, primarily formalizes rather than fundamentally extends these intuitions. Please see above weaknesses for more detailed concerns.

---

> ### Author Response · Authors · 2025-11-21
> **Response to Reviewer**
>
> We thank the reviewer for careful reading and valuable feedback. We note that many of the comments focus on the relationship between our contributions and prior work. Before addressing individual questions, we would like to clarify the positioning and core value of our paper, to avoid conflating our analysis with prior empirical or method-specific studies.
>
> Prior works on GR and DR have generally analyzed certain phenomena from the perspective of their own task settings or model designs. For example, DPR studies negative sampling in contrastive learning; DSI explores using model parameters as an index; some works point out embedding compression issues; others comment on training instability or degradation under scaling. However, these analyses are side observations tied to specific architectures, datasets, or training procedures. They do not constitute a unified or general theoretical account: they do not quantify the mathematical sources of DR’s limitations; they do not characterize the structural differences between GR and DR at the level of representational capacity or training objectives; and they do not offer a fair, controlled comparison of the two paradigms under the same assumptions. Therefore, our research goals and the depth of our conclusions are fundamentally different from these method-oriented works.
>
> Our paper aims to provide a systematic, fair, mechanism-level comparison of the two retrieval paradigms. We are the first to derive and formalize two structural bottlenecks intrinsic to DR: (1) calibration drift induced by local normalization (Theorem 3.1), and (2) a low-rank representational limit arising from bilinear interactions (Corollary 3.3). These results do not depend on any particular architecture or dataset, they capture properties of the DR paradigm itself. In parallel, through strict control of backbone models, training signals, corpus size, and the amount of trainable parameters, we construct a fair experimental environment in which the scaling behaviors of GR and DR can be cleanly observed. This allows us to isolate and validate their respective advantages and limitations.
>
> In summary, the novelty of our work lies in providing the first unified, systematic, and fair comparison of GR and DR as retrieval paradigms, going beyond the prior literature, which offers fragmented, setting-specific insight, and establishing a mechanism-level framework that we hope will serve as a foundation for future retrieval research.
>
>
> For W1:
> The reviewer notes that the limitations of DR: local normalization and low-rank bottlenecks, are intuitive and broadly recognized in the community. We fully agree. However, our contribution lies in providing the first unified mathematical characterization that formalizes these intuitions, revealing: where the inherent limitations of DR fundamentally originate; how these limitations are influenced by corpus scale and training dynamics; and why GR does not inherit these limitations.
>
> Our goal is not to claim that GR universally outperforms DR, but to theoretically explain the mechanism-level differences between the two paradigms, and to examine the potential advantages GR may possess as a consequence.
>
> For W2:
> The reviewer observes that Fig. 1 (more negatives lead to better performance) seems inconsistent with Table 1 (higher hard-negative ratio leads to worse performance). We clarify: Fig. 1 varies the number of negatives, primarily easy negatives; Table 1 varies the ratio of hard negatives, not the quantity of negatives.
>
> These manipulations are conceptually distinct. As stated in the paper, the purpose of Table 1 is not to evaluate the intrinsic value of hard negatives, but to demonstrate how sensitive DR is to the negative-sampling distribution. Because we did not filter hard-negative quality, increasing their ratio naturally introduces: excessive gradient noise, instability due to the absence of easy negatives, and model collapse when the hard-negative ratio approaches 1.Prior work has also reported similar instabilities when hard-negative sampling is overly aggressive [1].
>
> Thus, instead of contradicting Fig. 1, Table 1 reinforces our core message: Due to local normalization, DR’s optimization is highly fragile with respect to negative-sampling strategies.

---

> > ### Author Response · Authors · 2025-11-21
> > **Response to Reviewer (cont.)**
> >
> > For W3:
> > Pretrained model sizes are fixed, and the available model-size granularity is sparse (e.g., Qwen3 offers only 0.6B, 4B, and 8B below 10B), making it difficult to construct smooth scaling curves. To fairly compare GR and DR, both must share the same backbone; otherwise, scaling behavior is not comparable. Thus, we scale the adapter size to simulate changes in trainable parameter budget.
> >
> > To address the reviewer’s concern, we also conducted supplementary experiments using Qwen3-4B and 8B on NQ. The corresponding Hits@1 are:
> >
> >  DR: 0.6B: 61.4% → 4B: 62.9% → 8B: 63.6%
> >  GR: 0.6B: 63.8% → 4B: 66.5% → 8B: 68.1%
> >
> > Although the data points are limited, the trend is clear: GR benefits more from scaling than DR, consistent with our main conclusion.
> >
> > The reviewer notes that GR performs significantly worse than DR on MS MARCO, which appears inconsistent with the theoretical advantages we describe. We fully agree with this observation, and offer the following deeper explanation.
> >
> > The purpose of our paper is not to claim that GR is already SOTA across all settings. Instead, the goal is to show that GR does not inherit the two fundamental limitations of DR. GR’s current performance shortfall primarily arises from two causes:
> >
> > (1) GR is far more sensitive to data design and task priors than DR. Key GR design choices (such as codebook quantization algorithms, hierarchical depth, docid length, decoding constraints, etc.) have substantial impact on GR performance. In contrast, DR is relatively insensitive to these choices (which is indeed an advantage of DR). For fairness, we did not tune these design choices specifically for MS MARCO. For our purposes, the essential signal is the slope of the scaling curve, and the empirical slopes we observe align closely with our theoretical predictions, demonstrating GR’s underlying potential.
> >
> > (2) GR requires repeated exposure to documents, and MS MARCO’s distribution is inherently unfavorable. Training GR resembles training a memory system: the model must repeatedly absorb document information. However, MS MARCO exhibits low query-document coverage, limited overlap between training and validation documents, and sparse per-document supervision. This makes GR’s generalization to unseen documents particularly challenging. Some prior works have shown that with carefully optimized docid designs and better training signals, GR can indeed surpass DR on MS MARCO. Our contribution is to identify why GR has potential, even if that potential is not yet fully realized under all configurations.
> >
> >
> > For W4:
> > We appreciate the reviewer highlighting the importance of practical deployment concerns.
> > For latency, we conducted additional latency measurements on NQ across two corpus sizes (300k documents vs. 5M documents), comparing: DR-Flat (brute-force search), DR-ANN (HNSW index), and GR (constrained beam search). All results are normalized by GR latency (set to 1):
> >
> > | Corpus Size | DR-Flat | DR-ANN | GR |
> > | 300k        | 0.08    | 0.05   | 1  |
> > | 5M          | 0.31    | 0.09   | 1  |
> >
> > Although GR has higher absolute latency, DR-Flat’s latency grows sharply with corpus size, showing no clear advantage at large scale; DR-ANN significantly reduces latency, but introduces notable accuracy degradation; GR maintains stable latency regardless of corpus size, since constrained decoding does not depend on external index search.
> >
> > For inference computational cost, all methods in our comparison share the same model backbone, so the fundamental FLOPs per forward pass are identical. The main difference lies in autoregressive decoding vs. vector search. We have already analyzed this efficiency trade-off together with latency results.
> >
> > For storage requirements, DR must store a dense embedding vector for every document together with the associated ANN index structures. In contrast, GR only needs to store symbolic docids and a prefix trie (or equivalent lexicon) built over them, whose size is typically orders of magnitude smaller than a dense vector index for the same corpus.
> >
> >
> > For W5:
> > We thank the reviewer for highlighting that retrieval performance is influenced by many factors. The goal of this paper is to first work under a simplified and controlled setting, focusing on characterizing the structural bottlenecks of DR in terms of local normalization and low-rank representations. In the discussion section, we partially extend the analysis to how factors such as negative sampling strategies, docid design, and data distribution may amplify or mitigate these bottlenecks. Due to space and scope limitations, we do not provide empirical analysis for all such factors, which is a limitation of the current work.
> >
> > For W6:
> > We thank the reviewer for their attention to the proof details of Theorem 3.4. We will make the proof more rigorous in the final version to improve its overall mathematical precision and readability.

---

### Official Review · Reviewer_K22a · 2025-11-01

**Soundness:** 3
**Presentation:** 3
**Contribution:** 2
**Rating:** 2
**Confidence:** 4

**Summary:**

The paper theoretically and empirically investigate how GR fundamentally diverges from DR in both learning objectives and representational capacity.
The authors argue:
1. GR uses globally normalized maximum-likelihood training, unlike DR’s locally-normalized contrastive training.
2. GR scales better with larger corpora and models, avoiding optimization drift and low-rank bottlenecks present in DR.
They experiment over NQ and MSMARCO to validate these theoretical findings.

**Strengths:**

- The paper provides a well-structured and intuitive comparison between dense retrieval and generative retrieval, highlighting their distinct learning objectives and representational properties.
- The authors conduct comprehensive experiments to verify their theoretical claims.
- The authors propose several potential solutions to improve the practical performance of generative retrieval.

**Weaknesses:**

Many of the findings seem to have already been explored. Could you clarify the comparison with prior work?:
- The discussion on learning objectives in dense retrieval appears related to [1], while the analysis of representational capacity seems closely connected to [2]. In addition, the comparison of GR vs. DR with respect to embedding size resembles findings in [3].
- For model scaling, why did you vary adapter size rather than model size? Also, when it's on model size, the scaling effects for DR and GR have been explored in [4] and [5], respectively.
- The zero-shot GT setting in Section 4.4 appears related to [6]. It would be helpful to clarify how your setup differs.

[1] Karpukhin, Vladimir, et al. "Dense Passage Retrieval for Open-Domain Question Answering." EMNLP (1). 2020.

[2] Luan, Yi, et al. "Sparse, dense, and attentional representations for text retrieval." Transactions of the Association for Computational Linguistics 9 (2021): 329-345.

[3] Lee, Hyunji, et al. "Generative multi-hop retrieval." arXiv preprint arXiv:2204.13596 (2022).

[4] Ni, Jianmo, et al. "Sentence-t5: Scalable sentence encoders from pre-trained text-to-text models." arXiv preprint arXiv:2108.08877 (2021).

[5] Tay, Yi, et al. "Transformer memory as a differentiable search index." Advances in Neural Information Processing Systems 35 (2022): 21831-21843.

[6] Yu, Wenhao, et al. "Generate rather than retrieve: Large language models are strong context generators." arXiv preprint arXiv:2209.10063 (2022).

**Questions:**

- Do you expect the experimental results to generalize to larger model scales?
- While you argue that GR benefits from greater representational capacity, prior work has discussed the limited generalization of generative retrieval [1], which seems to be due to limited capacity. Could you share your thoughts on this?
- Prior work generally shows that hard negatives help training [2,3], but Table 1 reports performance degradation as the hard-negative ratio increases. Could you provide intuition for this?

[1] Mehta, Sanket Vaibhav, et al. "Dsi++: Updating transformer memory with new documents." arXiv preprint arXiv:2212.09744 (2022).

[2] Karpukhin, Vladimir, et al. "Dense Passage Retrieval for Open-Domain Question Answering." EMNLP (1). 2020.

[3] Izacard, Gautier, et al. "Unsupervised dense information retrieval with contrastive learning." arXiv preprint arXiv:2112.09118 (2021).

---

> ### Author Response · Authors · 2025-11-21
> **Response to Reviewer**
>
> We thank the reviewer for careful reading and valuable feedback. We note that many of the comments focus on the relationship between our contributions and prior work. Before addressing individual questions, we would like to clarify the positioning and core value of our paper, to avoid conflating our analysis with prior empirical or method-specific studies.
>
> Prior works on GR and DR have generally analyzed certain phenomena from the perspective of their own task settings or model designs. For example, DPR [1] studies negative sampling in contrastive learning; DSI [5] explores using model parameters as an index; some works point out embedding compression issues; others comment on training instability or degradation under scaling. However, these analyses are side observations tied to specific architectures, datasets, or training procedures. They do not constitute a unified or general theoretical account: they do not quantify the mathematical sources of DR’s limitations; they do not characterize the structural differences between GR and DR at the level of representational capacity or training objectives; and they do not offer a fair, controlled comparison of the two paradigms under the same assumptions. Therefore, our research goals and the depth of our conclusions are fundamentally different from these method-oriented works.
>
> Our paper aims to provide a systematic, fair, mechanism-level comparison of the two retrieval paradigms. We are the first to derive and formalize two structural bottlenecks intrinsic to DR: (1) calibration drift induced by local normalization (Theorem 3.1), and (2) a low-rank representational limit arising from bilinear interactions (Corollary 3.3). These results do not depend on any particular architecture or dataset, they capture properties of the DR paradigm itself. In parallel, through strict control of backbone models, training signals, corpus size, and the amount of trainable parameters, we construct a fair experimental environment in which the scaling behaviors of GR and DR can be cleanly observed. This allows us to isolate and validate their respective advantages and limitations.
>
> In summary, the novelty of our work lies in providing the first unified, systematic, and fair comparison of GR and DR as retrieval paradigms, going beyond the prior literature, which offers fragmented, setting-specific insight, and establishing a mechanism-level framework that we hope will serve as a foundation for future retrieval research.
>
> For W1:
> (1)Our work is the first to formally define DR’s locally normalized surrogate and derive a theoretical lower bound for the local–global normalization gap, which sharply contrasts with the globally normalized nature of GR. Prior work such as DPR [1] contains no such theoretical framework.
> (2)Existing studies do not analyze the provable rank bottleneck of DR. The works mentioned by the reviewer (e.g., [2]) focus on methodological discussions such as late-interaction architectures and embedding engineering, but they center on dense retrieval and do not provide a theoretical explanation for DR’s representational limitations, nor do they address GR’s universal approximation property.
> (3)The embedding bottleneck discussed in [3] is merely an empirical and qualitative observation about information compression in bi-encoder models for multi-hop retrieval, used to motivate their adoption of generative retrieval for that task. It does not provide any theoretical analysis of low-rank constraints or representational capacity limits, nor does it offer a fair or controlled comparison of representational ability.
>
> For W2:
> Works [4][5] examine scaling of DR alone (e.g., Sentence-T5) or scaling of GR alone (e.g., DSI), but none compares how GR and DR behave under the same parameter budget, nor do they explain scaling behaviors from the perspectives of rank constraints and global normalization. Therefore, these empirical scaling studies do not overlap with our theoretical or experimental contributions.
>
> For W3:
> [6]is not a generative retrieval (GR) task: the model generates natural-language context rather than docids, and does not apply constrained decoding. It is essentially a QA task rather than a retrieval task, so it is not comparable to our setting.
>
> Our zero-shot GR experiments use completely untrained retrieval models that rely solely on prompting and perform trie-constrained decoding to generate docids. This is fundamentally different from [6], which generates context rather than docids and does not involve constrained decoding, thus constituting a QA task rather than a retrieval task. Instead, our setting is more closely related to zeroGR [7] (appearing after our submission), though still more strict: zeroGR applies light instruction tuning to teach retrieval interfaces, whereas our zero-shot setup relies entirely on ICL without any tuning.

---

> > ### Author Response · Authors · 2025-11-21
> > **Response to Reviewer (cont.)**
> >
> > For Q1&W2:  Why scale adapters rather than model size? Will results generalize to larger scales?
> >
> > Scaling adapters instead of backbone size is motivated by practical and methodological considerations. Retrieval systems rarely deploy extremely large models (e.g., >10B) because latency is critical and training data is typically limited, making such large models prone to underfitting the retrieval task. Pretrained model sizes are fixed, and the available model-size granularity is sparse (e.g., Qwen3 offers only 0.6B, 4B, and 8B below 10B), making it difficult to construct smooth scaling curves. To fairly compare GR and DR, both must share the same backbone; otherwise, scaling behavior is not comparable. Thus, we scale the adapter size to simulate changes in trainable parameter budget. Our largest adapter (0.8B) already exceeds the base model size (0.6B), which makes the scaling meaningful. To address the reviewer’s concern, we also conducted supplementary experiments using Qwen3-4B and 8B on NQ. The corresponding Hits@1 are:
> >
> >  DR: 0.6B: 61.4% → 4B: 62.9% → 8B: 63.6%
> >  GR: 0.6B: 63.8% → 4B: 66.5% → 8B: 68.1%
> >
> > Although the data points are limited, the trend is clear: GR benefits more from scaling than DR, consistent with our main conclusion.
> >
> > For Q2:  Generalization capabilities and relation to model capacity
> >
> > Regarding the reviewer’s concern about GR’s ability to generalize, GR indeed struggles with unseen documents, especially when docids are numeric or codebook-based. However, GR is not incapable of generalization: prior studies [8][9] demonstrate that when using textual or semantic docids, models can generalize to unseen documents via language-model reasoning.
> >
> > As we noted in the discussion section, continual learning is a central challenge for generative retrieval, and existing work has explored solutions [10][11]. Our paper proposes a hybrid solution that alleviates continual-learning issues: GR only generates coarse categories, DR conducts fine-grained retrieval within the predicted category, and new documents simply attach to category nodes without retraining GR. Similar strategies have been used in industrial systems [12].
> >
> > Overall, GR’s generalization and continual-learning capabilities depend more on data distribution, docid design, and training regime than on model size alone.
> >
> > For Q3:  Why performance decreases when increasing the hard-negative ratio?
> >
> > The purpose of the experiment was not to evaluate the benefit of hard negatives, but to demonstrate that DR is highly sensitive to negative-sampling strategies. Because we did not carefully filter negative samples, a high proportion of hard negatives degraded DR performance. This behavior is expected: Low-quality or overly difficult hard negatives introduce gradient noise. When the hard-negative proportion becomes too high (maximum = 1 in our experiment), the model lacks “easy” negatives, causing instability. Similar findings have been reported in prior work [13]. Thus, the observed degradation supports our key claim: due to local normalization, DR’s optimization quality heavily depends on the negative-sampling strategy.
> >
> > [1] Karpukhin, Vladimir, et al. "Dense Passage Retrieval for Open-Domain Question Answering." EMNLP (1). 2020.
> >
> > [2] Luan, Yi, et al. "Sparse, dense, and attentional representations for text retrieval." Transactions of the Association for Computational Linguistics 9 (2021): 329-345.
> >
> > [3] Lee, Hyunji, et al. "Generative multi-hop retrieval." arXiv preprint arXiv:2204.13596 (2022).
> >
> > [4] Ni, Jianmo, et al. "Sentence-t5: Scalable sentence encoders from pre-trained text-to-text models." arXiv preprint arXiv:2108.08877 (2021).
> >
> > [5] Tay, Yi, et al. "Transformer memory as a differentiable search index." Advances in Neural Information Processing Systems 35 (2022): 21831-21843.
> >
> > [6] Yu, Wenhao, et al. "Generate rather than retrieve: Large language models are strong context generators." arXiv preprint arXiv:2209.10063 (2022).
> >
> > [7] Sun, Weiwei, et al. "ZeroGR: A Generalizable and Scalable Framework for Zero-Shot Generative Retrieval." arXiv preprint arXiv:2510.10419 (2025).
> >
> > [8] Lee, Sunkyung, Minjin Choi, and Jongwuk Lee. "GLEN: Generative retrieval via lexical index learning." arXiv preprint arXiv:2311.03057 (2023).
> >
> > [9] Sun, Weiwei, et al. "Learning to tokenize for generative retrieval." Advances in Neural Information Processing Systems 36 (2023): 46345-46361.
> >
> > [10] Mehta, Sanket Vaibhav, et al. "Dsi++: Updating transformer memory with new documents." arXiv preprint arXiv:2212.09744 (2022).
> >
> > [11] Kishore, Varsha, et al. "Incdsi: Incrementally updatable document retrieval." International conference on machine learning. PMLR, 2023.
> >
> > [12] Guo, Jiafeng, et al. "Retrieval-in-the-Chain: Bootstrapping Large Language Models for Generative Retrieval." arXiv preprint arXiv:2510.13095 (2025).
> >
> > [13] Xiong, Lee, et al. "Approximate nearest neighbor negative contrastive learning for dense text retrieval." arXiv preprint arXiv:2007.00808 (2020).

---

> > > ### Comment · Reviewer_K22a · 2025-11-27
> > >
> > > Thank you for the detailed response.
> > >
> > > W1. Regarding the authors’ comment that *“The works mentioned by the reviewer (e.g., [2]) focus on methodological discussions such as late-interaction architectures and embedding engineering, but they center on dense retrieval and do not provide a theoretical explanation for DR’s representational limitations, nor do they address GR’s universal approximation property”*. I agree that these works do not analyze GR’s properties. However, I believe that [2] does provide a theoretical explanation for DR’s representational limitations, including those related to embedding dimensionality and document length. Could you please check this point again?
> > >
> > > W3. While [6] is not explicitly designed for retrieval tasks, its framework can be readily extended to such applications in a zero-shot setting.
> > >
> > > Q3. I understand the authors’ intention, but in practical settings, negative sampling is typically handled carefully. Even simple approaches, such as in-batch negatives have been shown to provide meaningful improvements. Therefore, I am not sure if the observations presented fully align with real-world use cases.

---

> > > > ### Author Response · Authors · 2025-11-28
> > > >
> > > > Thank you for your prompt reading and response.
> > > >
> > > > W1:
> > > > We thank the reviewer for the clarification. We agree that [2] indeed provides a theoretical analysis of dual-encoder capacity, especially by relating encoding dimension, document length, and the margin between positive and negative documents to derive lower bounds on the dimensionality required to separate documents.
> > > >
> > > > However, we view our analysis as complementary rather than redundant with [2]:
> > > > [2] focuses on how large the encoding dimension must be for a dual encoder to separate documents in the training set under specific assumptions on margins and document length. In contrast, our work characterizes a low-rank bottleneck of the relevance operator itself (Corollary 3.3), proving that under bilinear interactions, the induced relevance function is necessarily confined to a low-rank subspace. This conclusion is independent of any particular training sample configuration or margin assumptions, and is closer to a paradigm-level structural limitation. In addition, [2] does not connect these capacity results to local vs. global normalization, nor does it compare DR and GR within a unified framework.
> > > >
> > > > We will revise the paper to more explicitly incorporate [2] in the related-work discussion and to position our rank-based analysis as a paradigm-level complement, rather than a replacement.
> > > >
> > > > W2:
> > > > We believe that directly extending [6] to zero-shot generative retrieval is non-trivial, and its experimental conclusions cannot be directly carried over to the generative retrieval setting.
> > > >
> > > > Concretely, retrieval requires precisely locating specific entities or documents, which is fundamentally different from freely generating arbitrary context as in [6]. In a retrieval setting, if the output space is left unconstrained, the model will often produce drifted outputs: strings that do not belong to the candidate set and thus cannot be mapped to real documents. To avoid this, generative retrieval must impose constrained decoding that restricts the output space to valid docids.
> > > >
> > > > However, once such constraints are imposed, the model’s output space and its expression patterns are forcibly altered: the constrained output format often deviates substantially from the natural-language distribution encountered during pretraining. As a result, the zero-shot performance reported under unconstrained generation in [6] cannot be directly interpreted as an upper bound for constrained docid generation.
> > > >
> > > > Our zero-shot GR experiments, which show relatively low performance under strict constrained decoding, precisely validate this point: On one hand, they indicate that directly carrying over conclusions from free generation to constrained generation is overly optimistic; On the other hand, they also show that even under strong constraints, zero-shot generative retrieval remains non-trivially effective, which indirectly confirms that large language models have indeed acquired rich, retrieval-relevant knowledge during pretraining.
> > > >
> > > > We believe this highlights an important direction for future work: how to unlock and amplify the inherent retrieval capability of pretrained generative models under constrained docid generation.
> > > >
> > > >
> > > > Q3:
> > > > We thank the reviewer for the further remark and agree that, in real-world systems, negative sampling is typically designed with great care. Our intention in Table 1 is not to suggest that practical systems would deliberately adopt poor negative-sampling strategies, but rather to highlight a mechanism-level fact: Due to local normalization, the optimization quality of DR is structurally sensitive to the negative-sample distribution itself.
> > > >
> > > > When we only change the ratio of hard negatives, without redesigning mining or sampling strategies, the observed performance drop directly reflects this sensitivity. In real systems, more sophisticated negative-sampling designs can certainly mitigate the issue, but they cannot remove DR’s fundamental dependence on the negative distribution.
> > > >
> > > > We will clarify the purpose of this experiment more explicitly in the final version.

---

### Note · Authors · 2025-12-08

I have read and agree with the venue's withdrawal policy on behalf of myself and my co-authors.